# Initiation of lumen formation from junctions via differential actomyosin contractility regulated by dynamic recruitment of Rasip1

Jianmin Yin [1,5] ✉, Niels Schellinx[1], Ludovico Maggi [1], Kathrin Gundel[1,3], Cora Wiesner[1], Maria Paraskevi Kotini[1], Minkyoung Lee[1,4], Li-Kun Phng [2], Heinz-Georg Belting [1,5] ✉ & Markus Affolter [1,5] ✉

De novo lumen formation necessitates the precise segregation of junctional proteins from apical surfaces, yet the underlying mechanisms remain unclear. Using a zebrafish model, we develop a series of molecular reporters, photoconvertible and optogenetic tools to study the establishment of apical domains. Our study identifies Rasip1 as one of the earliest apical proteins recruited, which suppresses actomyosin contractility at junctional patches by inhibiting NMII, thereby allowing for the sustained outward flow of junctional complexes. Following the establishment of apical compartments, Rasip1 shuttles between junctions and the apical compartments in response to local high tension. Rasip1 confines Cdh5 to junctions by suppressing apical contractility. Conversely, the recruitment of Rasip1 to junctions is regulated by Heg1 and Krit1 to modulate contractility along junctions. Overall, de novo lumen formation and maintenance depend on the precise control of contractility within apical compartments and junctions, orchestrated by the dynamic recruitment of Rasip1.

Endothelial cells (ECs) polarize and generate apical domains and lumens to create a functional, multicellular tubular network of blood vessels. Cell-cell contacts, formed by cadherin-based adherens junctions (AJs) and tight junctions, reside between the apical and basolateral membranes and seal the endothelial barrier. To establish appropriate de novo apical compartments, ECs must regulate the proper segregation of junctional proteins from apical surfaces. However, the mechanisms governing the establishment and maintenance of distinct cellular compartments in vivo remain poorly understood. Obtaining a precise map of protein recruitment and segregation during apical polarization, and a comprehensive understanding of this process, has proven to be extremely difficult in vivo due to the lack of visualizable systems, proper reporters, and manipulative tools.

The formation of a blood vessel network during embryogenesis requires both vasculogenesis and angiogenesis. Individual EC progenitors aggregate into cord structures that open central cavities during vasculogenesis[1,2]. Later during development, new vessels are primarily formed through angiogenesis, a process in which new vessels sprout from pre-existing ones, grow, and connect to adjacent sprouts or other vessels[3–5]. Many cellular and molecular commonalities in forming de novo lumens have been revealed from studies in various contexts with different model organisms[6,7]. In this study, we used the formation of the dorsal longitudinal anastomotic vessel (DLAV) in zebrafish as an experimental model to analyze the establishment of apical compartments in vivo[8]. During DLAV formation, a new vessel is built by the interaction of two approaching tip cells[3]. Upon contact between the two tip cells, AJs proteins, such as Cdh5, are deposited at

[1]Department of Cell Biology, Biozentrum, University of Basel, Basel, Switzerland. [2]Laboratory for Vascular Morphogenesis, RIKEN Center for Biosystems Dynamics Research, Kobe, Japan. [3]Present address: Universitätsklinikum Bonn, Bonn, Germany. [4]Present address: Department of Biosystems Science and Engineering, ETH Zürich, Basel, Switzerland. [5]These authors jointly supervised this work: Jianmin Yin, Heinz-Georg Belting, Markus Affolter. ✉e-mail: jianmin.yin@unibas.ch; heinz-georg.belting@unibas.ch; markus.affolter@unibas.ch

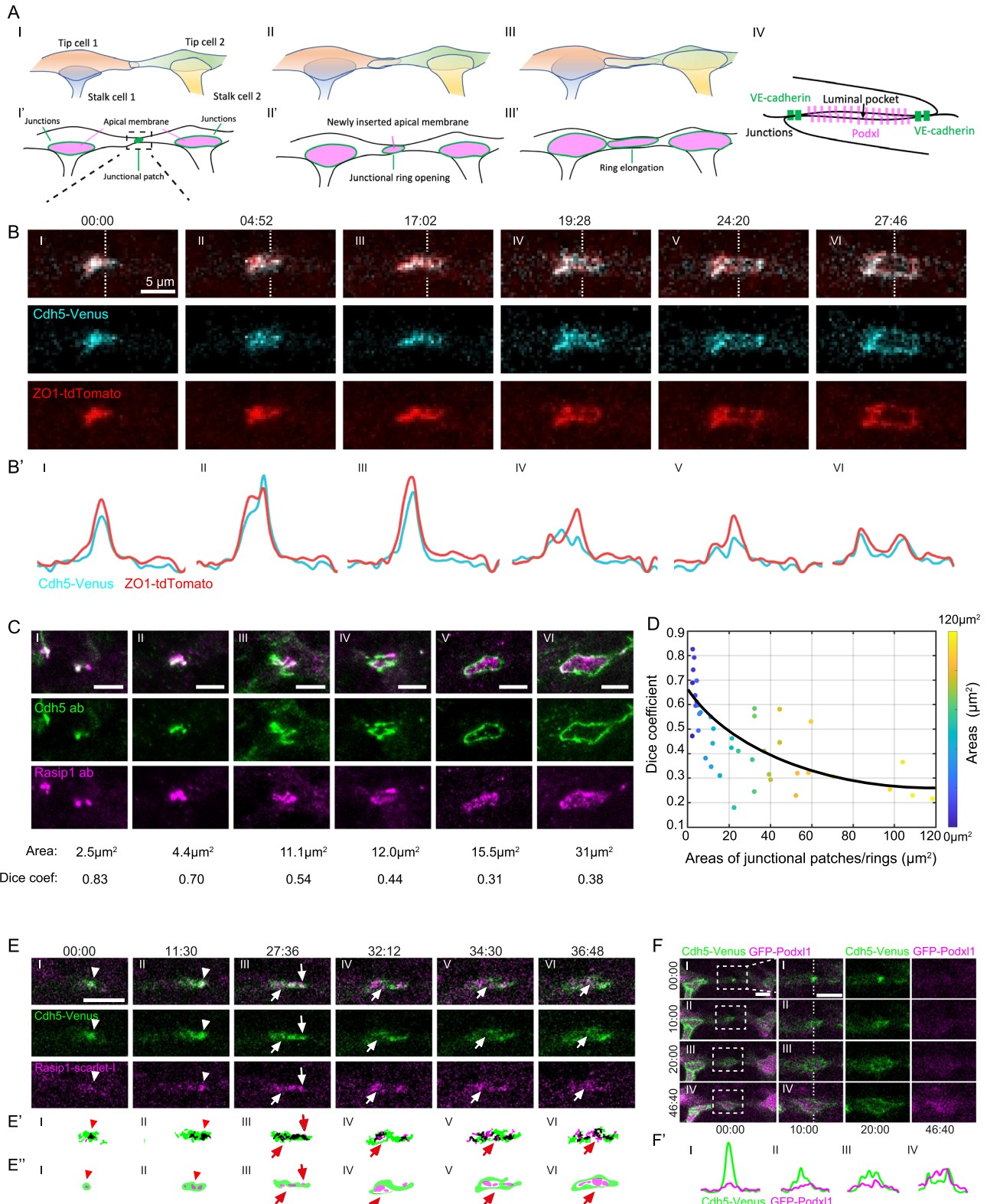

the contact site (Fig. 1A)[9,10]. At these adhesion sites, cells deposit de novo apical membranes while the initial adhesion spots open into junctional rings with apical membrane in between (Fig. 1A and Supplementary Movie 1)[3]. Here, the key questions we aim to address are how different cellular compartments are established and maintained during de novo lumen formation.

The actin cytoskeleton and its regulatory Rho GTPases are implicated in EC polarization and junctional remodeling[11–13]. Rasip1

(Ras-interacting protein 1) is a potent regulator of Rho GTPase activity[14–17]. Previous studies in mice and zebrafish suggest that Rasip1 is required for the clearance of apical junctions during vasculogenesis and angiogenesis. In mouse mutant embryos, the angioblasts fail to localize junctional proteins to the cord periphery during vasculogenesis of the dorsal aorta (DA)[15,18,19]. Similarly, in zebrafish embryos undergoing DLAV formation, *rasip1* mutants manifest ectopic and reticulated junctional patterns within the presumed apical

**Fig. 1 | Rasip1 is dynamically involved in the establishment of apical compartments. A** Schematic depicting the anastomosis of two tip cells in the formation of the DLAV. A stable cell-cell contact site is formed with the deposition of junctional proteins, including Cdh5 (I). The cell-cell contact site opens into a ring structure while the apical membrane is inserted into the luminal pocket within the junctional ring (II). Cell rearrangement leads to the elongation of the junctional ring (III). A transverse view of the luminal pocket located within the junctional ring (IV). **B** Time-lapse imaging of Cdh5-Venus and ZO1-tdTomato, captured from 30 hpf throughout the patch-to-ring transition. (**B'**) Intensities of Cdh5-Venus and ZO1-tdTomato along the dashed lines in (**B**). **C** Antibody staining for Cdh5 and Rasip1 at junctional patches and rings of varying sizes, from initial cell-cell contacts to the formation of stable junctional rings. The Dice-Sørensen coefficient was calculated as twice the joint area of the two signals divided by the sum of the area of each signal after automatic thresholding. **D** The Dice coefficient and the corresponding area of various junctional patches and rings, with data points color-coded by area (*n* = 42 cells from 12 embryos). **E** Time-lapse images showing the recruitment of Rasip1-Scarlet-I to the junctional patches and nascent junctional rings. Arrowheads mark Rasip1 clusters within the junctional patches. Arrows mark Rasip1 clusters localized at the periphery of junctional patches or within the nascent apical compartments. (**E'**) Automatic thresholding of Cdh5-Venus and Rasip1-Scarlet-I in (**E**). Black corresponds to the colocalized signals after thresholding. (**E"**) Graphic diagrams showing the localizations of signals in (**E'**). Consistent observations were made in 6 samples from 4 independent experiments. **F** Time-lapse images showing the recruitment of GFP-Podxl1 to the apical compartments. (**F'**) Intensities of Cdh5-Venus and GFP-Podxl1 along the dashed lines in (**F**). Consistent observations were made in 8 samples from 4 independent experiments. All scale bars: 5 μm. Source data are provided as a Source Data file.

compartments[20]. Rasip1 has been shown to inhibit RhoA activity by binding to the GTPase-activating protein Arhgap29[18,19,21]. A mechanism has been proposed that Rasip1 clears apical junctions via the mechanical sliding of junctional proteins along actin tracks to the cord periphery through activating Cdc42 and nonmuscle myosin II (NMII) during vasculogenesis of the DA[14]. However, direct visualization of this process, how Rasip1 possibly switches between activator and inhibitor of NMII, and how the apical and junctional contractilities are coordinated during de novo lumen formation remain elusive.

In this work, to gain a comprehensive understanding of apical clearance, we explore both the temporal and spatial dynamics of apical compartmentalization and junctional remodeling during de novo lumen formation. By developing and utilizing a series of molecular reporters, we identify Rasip1 as the earliest apical protein recruited to the initial cell-cell contact sites, segregating with Cdh5 during de novo lumen formation. Further, after the establishment of the apical compartment, Rasip1 shuttles between junctions and the apical compartments to high-tension sites. Using high-resolution, multi-dimensional live imaging, photoconversion, pharmacological inhibition, and optogenetic tools, we demonstrate that Rasip1 initiates the sustained outward flow of junctional complexes and further stabilizes them to junctions by inhibiting apical contractility and constraining Cdh5 detachment.

## Results

### Junction remodeling and de novo lumen formation in the DLAV
Morphogenesis of the DLAV in the zebrafish is characterized by the de novo formation of a lumen between two neighboring sprouts at around 30 hours post-fertilization (hpf). Here, luminal pockets form an apical compartment, which later connects to neighboring lumens to generate vascular patency (Fig. 1A). To elucidate the emergence of a de novo apical compartment from an initial cell-cell contact site between two tip cells, we employed the Cdh5 reporter, Cdh5-Venus [*Tg(cdh5:cdh5-TFP-TENS-Venus)^uq11bh*], and the ZO-1 (tjp1a) reporter, [*TgKI(tjp1a-tdTomato)^pd1224*][22,23]. The two junctional markers strongly colocalized and exhibited a similar rearrangement pattern, transitioning from junctional patches to junctional rings within an average 30-minute time window (Fig. 1B and Supplementary Movie 2). The initial cell-cell contact sites expanded toward the periphery, forming junctional patches (Fig. 1B(I-III)). These junctional patches became hollow inside and transitioned into nascent junctional rings (Fig. 1B(V)). Subsequently, the junctional rings further expanded, generating de novo apical compartments with minimal junctional material inside (Fig. 1B(VI)).

We next explored the recruitment of apical proteins during the initial lumen formation of the DLAV. Rasip1, a vascular-specific regulator of GTPases, has been shown to localize strongly at the apical compartments[20]. We performed immunostaining to visualize Cdh5 and Rasip1 at junctional patches and rings over the time course of anastomotic lumen formation, from initial cell-cell contacts to the formation of stable junctional rings between tip cells (Fig. 1C). Using the

Dice-Sørensen coefficient, calculated as twice the joint area of the two signals divided by the sum of the area of each signal after automatic thresholding, we quantified the size of the junctional patches/rings and the degree of colocalization between the two signals (Fig. 1D). Our analysis reveals a transition from strong colocalization at initial contacts to segregation in junctional rings, with Rasip1 localized centrally and Cdh5 positioned at the periphery (Fig. 1C, D).

To further investigate the dynamic recruitment and segregation of these proteins, we generated a fluorescent reporter line, [*Tg(fli1a: Rasip1-Scarlet-I)^ubs59*], to follow the distribution of Rasip1 during lumen formation in the DLAV. In agreement with the immunofluorescent analyses, Rasip1-Scarlet-I was strongly recruited to the initial contact sites (Fig. 1E(I and II), S1A, B, and Supplementary Movie 3). During the patch-to-ring transition, Rasip1 began segregating from Cdh5 by translocating to the periphery as clusters (Fig. 1E(III and IV), S1A(I-III), and S1B(I-III)). Further segregation of Rasip1 and Cdh5 resulted in spatial partitioning, with Cdh5 localized at the periphery and Rasip1 concentrated in the central region (Fig. 1E(IV-VI), S1A(IV and V), and S1B(IV and V)). After the patch-to-ring transition, most of the Rasip1 protein was found inside the junctional rings at the presumed apical membrane compartment (Fig. 1E(VI), S1A(V), and S1B(V)). In contrast, the surface glycoprotein Podocalyxin (PODXL) appeared in the nascent junctional rings after the patch-to-ring transition, as visualized by the Podxl1 live reporter [*Tg(fli1a: GFP-Podxl1)^ncv530Tg*] and antibody staining (Fig. 1F, S1C, and S1D)[24]. The direct comparison between Rasip1 and Podxl1 further demonstrated that during DLAV anastomosis, Rasip1 was recruited to the prospective apical membrane prior to Podocalyxin (Fig. S1E).

### Poor maintenance of apical compartments in *rasip1* mutants
Previous studies have demonstrated that Rasip1 is required for the removal of junctional complexes during the establishment of the apical membrane compartment[14,20]. To explore the mechanisms by which Rasip1 controls this clearing of junctional material, we first analyzed the temporal and spatial dynamics of this process in both wild-type embryos and *rasip1* null mutants[20]. We examined the shape of junctions via antibody staining for ZO-1 and Cdh5 during DLAV formation. In wild-type embryos, junctions exhibited distinct ring-like structures characterized by well-defined boundaries and predominantly cleared apical compartments, with little presence of junctional proteins (Fig. 2A(I)). In contrast, *rasip1* mutants displayed ectopic clusters of Cdh5 and ZO-1 (Fig. 2A(II), white arrowheads) or linear-shaped junctions (Fig. 2A(II), green arrowheads) within the apical compartments. The boundaries of the presumed apical compartments appeared fuzzy and discontinuous (Fig. 2A(II), white arrows). We observed similar phenotypes with the Cdh5-Venus live reporter in *rasip1* mutants (Fig. 2B(I and II)). These ectopic junctions were observed both within the newly formed apical compartments between tip cells and the pre-existing apical domains between tip and stalk cells, and were referred to as reticulated junctions (Fig. 2B(I and II)). By measuring the average

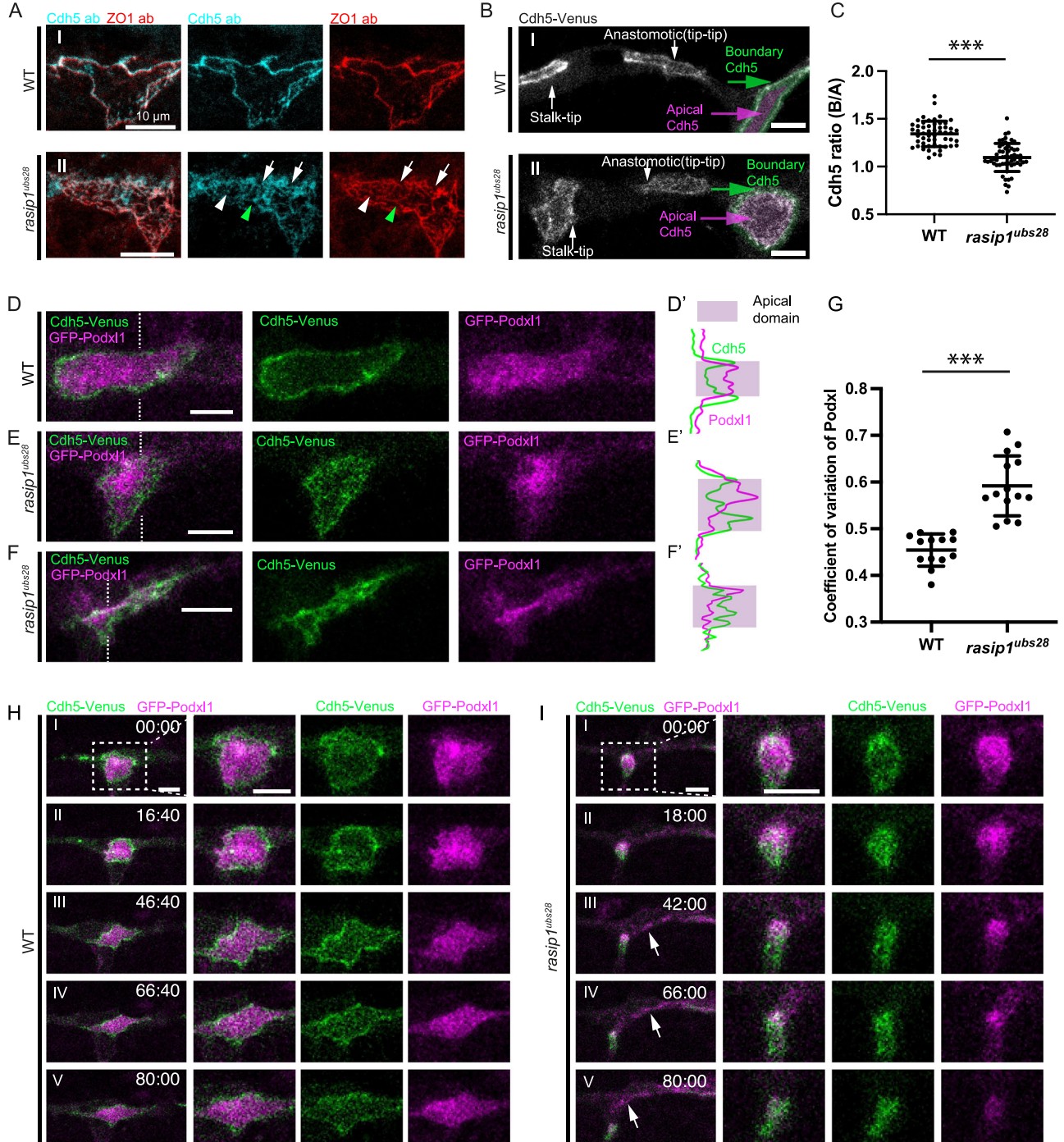

**Fig. 2 | Apical clearance and lumenal defects in *rasip1* mutants. A** Antibody staining for Cdh5 and ZO-1 in wild-type embryos and *rasip1* mutants at 32 hpf. White arrows indicate discontinuous junctions. White and green arrowheads indicate junctional clusters and linear junctional structures within the presumed apical compartments, respectively. **B** Expression of Cdh5-Venus in wild-type embryos and *rasip1* mutants at 32 hpf. Green and magenta masks label the boundary regions (junctions) and the presumed apical compartments, respectively. **C** Quantification of the ratio of the average Cdh5 intensity between the boundary regions (green masks in (**B**)) and the apical compartments (magenta masks in (**B**)), hereafter referred to as the boundary-to-apical (B/A) ratio. (WT: *n* = 53 cells from 10 embryos;

*rasip1*: *n* = 55 cells from 11 embryos). Data are presented as mean ± SD. ***P < 0.0001, 2-tailed unpaired t-test. **D–F** Cdh5-Venus and GFP-Podxl1 in wild-type embryos (**D**) and *rasip1* mutants (**E**, **F**). (**D'–F'**) Intensities of Cdh5-Venus and GFP-Podxl1 along the dashed lines in (**D–F**). **G** Coefficient of variation of Podxl1 per apical compartment, presented as mean ± SD (WT: *n* = 13 cells from 6 embryos; *rasip1*: *n* = 14 cells from 7 embryos). ***P < 0.0001, 2-tailed unpaired t-test. **H, I** Time-lapse images of Cdh5-Venus and GFP-Podxl1 in wild-type embryos (**H**) and *rasip1* mutants (**I**). White arrows in (**I**) indicates GFP-Podxl1 that emerged outside of junctions. Consistent observations in over 10 samples from 4 independent experiments per group. All scale bars: 10 μm. Source data are provided as a Source Data file.

levels of Cdh5-Venus within the boundary areas (Fig. 2B, green masks) as compared to the apical compartments (Fig. 2B, magenta masks), we confirmed the ectopic presence of junctional proteins within the apical compartments in *rasip1* mutants (Fig. 2C).

To determine whether the apical clearance defects in *rasip1* mutants are linked to defects in apical polarization, we performed live imaging of GFP-Podxl1 in both wild-type embryos and *rasip1* mutants. In wild-type embryos, Podxl1 displayed a largely uniform distribution

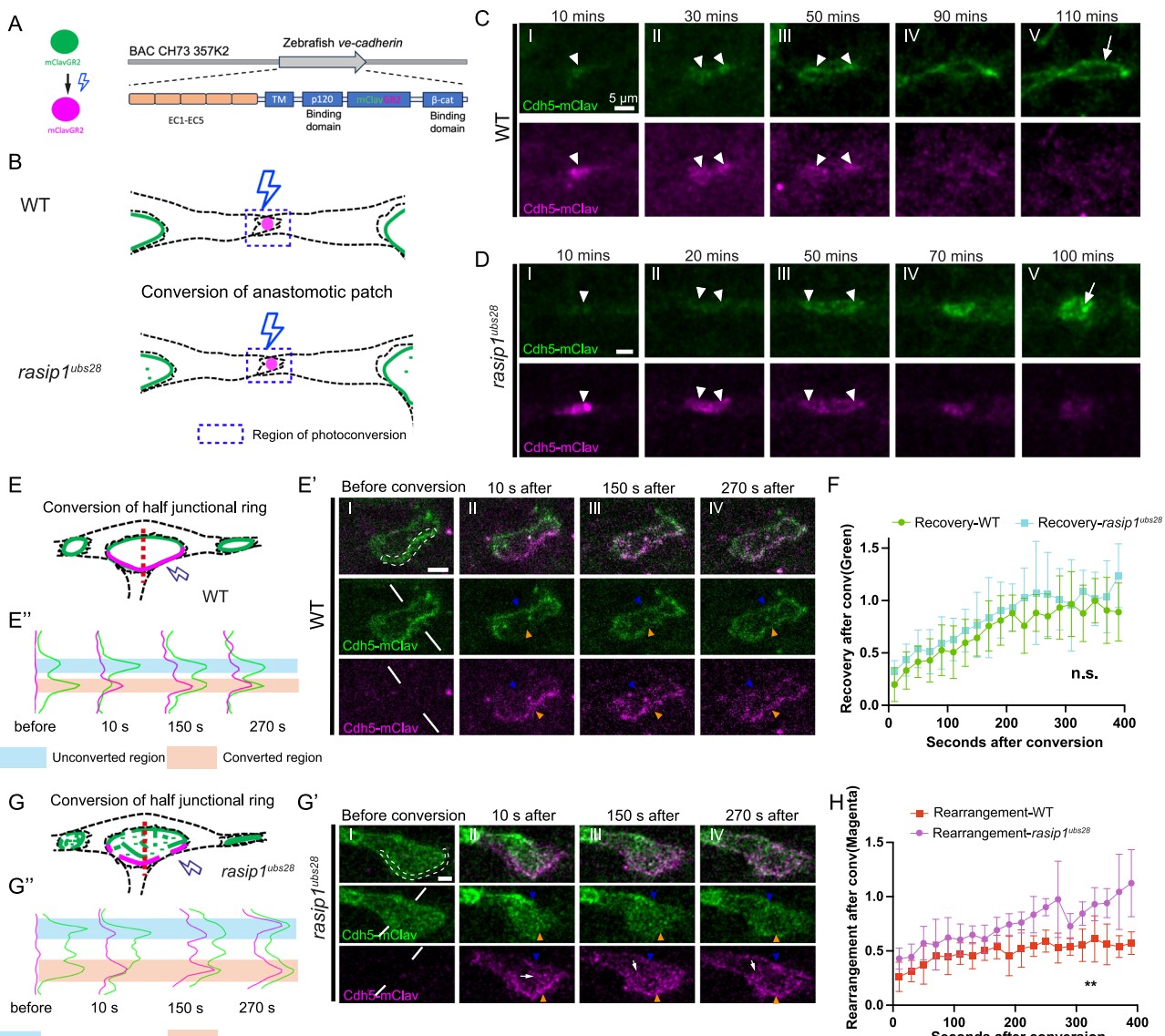

**Fig. 3 | Cdh5 is poorly restricted to the junctions in *rasip1* mutants. A** Schematic drawing of the photo-conversion of mClavGR2 and *cdh5-mClav* cDNA recombined into the *cdh5* BAC clone. **B** Diagrams of photo-conversions at the initial cell-cell contact sites (anastomotic patches) in wild-type embryos and in *rasip1* mutants. **C**, **D** Time-lapse imaging of Cdh5-mClav after photo-conversion specifically at junctional patches. White arrowheads mark the converted Cdh5-mClav. White arrows indicate the junctional ring and reticulated junctions formed in wild-type embryos and *rasip1* mutants, respectively. Consistent observations were made in over 6 samples from 3 independent experiments per group. **E**–**H** Photo-conversions on half junctional rings as depicted in diagrams for wild-type embryos (**E**) and equivalent regions in *rasip1* mutants (**G**). (**E'**, **G'**) Time-lapse imaging of Cdh5-mClav after photo-conversion of half junctional rings in wild-type embryos (**E'**) and equivalent regions in *rasip1* mutants (**G'**). White dashed lines mark the converted region before conversion. Blue and orange arrowheads indicate the unconverted and converted half rings, respectively. White arrows indicate converted Cdh5 in the apical domain. (**E''**, **G''**) Intensities of converted and unconverted Cdh5-mClav along the white lines in (**E'**) and (**G'**), corresponding to the red dashed lines in the diagrams in (**E**) and (**G**). **F** The relative intensity of green Cdh5-mClav in the converted half rings compared to the unconverted half rings in wild-type embryos (*n* = 5 experiments) and *rasip1* mutants (*n* = 6 experiments), presented as mean ± SD. N.s., not significant, 2-tailed unpaired t-test on the area under the curve. **H** The relative intensity of magenta Cdh5-mClav in the unconverted half rings compared to the converted half rings in wild-type embryos (*n* = 5 experiments) and *rasip1* mutants (*n* = 6 experiments), presented as mean ± SD. **P = 0.0028, 2-tailed unpaired t-test on the area under the curve. All scale bars: 5 µm. Source data are provided as a Source Data file.

inside the junctional rings (Fig. 2D, D'). In *rasip1* mutants, Podxl1 exhibited an uneven distribution, with aggregates forming between reticulated junctions (Fig. 2E, E', F, F'). The overall signal appeared elevated in the center (Fig. 2E, E') or at the periphery of the apical compartments (Fig. 2F, F') in the mutants. Furthermore, the GFP-Podxl1 signal within the apical compartments of *rasip1* mutants was more variable when compared to wild-type embryos (Fig. 2G). Time-lapse imaging of Cdh5-Venus and GFP-Podxl1 revealed that while Podxl1 remained confined within junctional rings in wild-type embryos (Fig. 2H), it progressively dispersed from the apical compartment in

*rasip1* mutants, with some accumulation outside the junctional ring (Fig. 2I and Supplementary Movie 4). Taken together, these data indicate that the initial apical polarization in *rasip1* mutants remains largely unaffected, as suggested by the recruitment of Podxl1. However, the proper localization of Podxl1 within the apical domain, as well as Cdh5 localization to junctions, is poorly maintained in *rasip1* mutants.

### Insufficient restriction on junctional Cdh5 in *rasip1* mutants
The above findings show that the loss of Rasip1 function not only affects apical clearance of junctional proteins but also impacts the

proper separation of compartment boundaries at the cell-cell junctions. To elucidate a possible connection between apical clearance and junction formation, we generated a photoconvertible Cdh5 reporter, Cdh5-mClav [*TgBAC(cdh5:cdh5-mClavGR2)^ubsS8*] (Fig. 3A), which allowed us to track the dynamics of different pools of Cdh5 during apical polarization and junctional ring formation. Different pools of Cdh5 were labeled by photo-conversion of Cdh5-mClav in discrete regions of interest (ROIs) at different stages of anastomosis. We first selectively photoconverted entire junctional patches between two tip cells, as indicated by the squares drawn around them (Fig. 3B). Upon complete conversion, we monitored subsequent junctional remodeling in both wild-type embryos and *rasip1* mutants (Fig. 3C, D). The photo-converted Cdh5 from the junctional patches rearranged towards the periphery and eventually became part of junctional rings together with newly incorporated green Cdh5, both in wild-type and *rasip1* mutant embryos (Fig. 3C(III), 3D(III), arrowheads). The converted Cdh5 slowly faded during subsequent junctional remodeling, possibly due to recycling and photo-bleaching in both conditions (Fig. 3C(V), D(V)). Interestingly, and in sharp contrast to wild-type embryos with elongating junctions, the junctional ring in *rasip1* mutants quickly transformed into a reticulated pattern, losing clear boundaries and displaying ectopic junctional complexes within the presumed apical domain (Fig. 3D(V), arrow). Further time-lapse analyses of *rasip1* mutants revealed that a large portion of the initial adhesion sites transitioned into a ring shape temporarily ($n = 24/35$ cells from 10 embryos) but subsequently lost their defined localization ($n = 22/24$ cells from 10 embryos) (Fig. S2). These observations suggest that Rasip1 stabilizes Cdh5 at the cell junctions during the formation and enlargement of the junctional ring.

To analyze the dynamics of Cdh5 at the junction in more detail, we selectively photo-converted half of the pre-existing junctional rings between stalk and tip cells with hand-drawn ROIs in wild-type embryos and in *rasip1* mutants (Fig. 3E–H and Supplementary Movies 5, 6). Exposure to 405 nm laser light within the ROI quantitatively converted Cdh5-mClav at the junction (shown in magenta), while the other half of the ring remained unconverted (shown in green) (Fig. 3E', G'). We first quantified the recovery speed of green Cdh5-mClav in both wild-type and mutant embryos. In both conditions, the converted half-rings took approximately 200 seconds to reach 90% of the green fluorescence intensity measured in the unconverted half-ring, suggesting a rapid incorporation of new junctional proteins in both conditions (Fig. 3F). Conversely, we also assessed to what extent the converted Cdh5 moved away from the converted region. In wild-type embryos, the converted half-ring exhibited a magenta Cdh5 level two times higher than that of the unconverted half-ring even after 400 seconds (Fig. 3E'(IV), E", H). In contrast, a large amount of converted Cdh5 appeared in the apical domain as clusters and in the unconverted half-ring in *rasip1* mutants (Fig. 3G'(II-IV), G"). The unconverted half-ring displayed a comparable level of magenta Cdh5-mClav approximately 200-300 seconds after conversion (Fig. 3H). Thus, in *rasip1* mutants, junctional Cdh5 shows higher mobility and is less restricted, indicating that Rasip1 is required for the stabilization of junctions.

### Increased mobility of Cdh5 clusters in *rasip1* mutants

The above observations indicate that Rasip1 is essential for stabilizing Cdh5 at cell-cell junctions. In the absence of Rasip1, Cdh5 exhibits increased mobility toward the apical compartment. To further analyze this, we tracked individual Cdh5 particles at junctions and within the apical compartment. In wild-type embryos, only a few Cdh5-Venus particles appeared in the apical compartments (Fig. 4A, magenta arrowheads, and Supplementary Movie 7). However, in *rasip1* mutants, a greater number of Cdh5 clusters or linear fragments detached from junction boundaries and moved toward the apical domain (Fig. 4B, C, magenta arrowheads, and Supplementary Movies 8, 9). Notably, new Cdh5 was deposited at junctional boundaries, while fragments of old junctional clusters detached and moved toward the apical domain (Fig. 4B, C, green arrowheads). This detachment was further validated by tracking photo-converted Cdh5 particles, confirming their detachment from junctions into the apical domain (Fig. S3A, arrows).

To quantify these junction dynamics, we employed Particle Image Velocimetry (PIV) on high-temporal and spatial-resolution movies, which allowed us to track and measure Cdh5 movements both at the apical membrane and junctional boundaries (Fig. 4A(VI) −C(VI))[25]. In *rasip1* mutants, we detected significant inward movements toward the apical compartments from both the boundary and apical regions. Velocity maps were generated from frames captured at 36-second intervals using PIV and combined into a heat map displaying total particle displacement over 360 seconds in the radial direction. Different colors in the map represent velocities relative to the center (Fig. 4D, E). In wild-type embryos, movements were primarily lateral, correlating with the outward expansion of junctional rings (Fig. 4D). Conversely, in *rasip1* mutants, significant inward (blue) movements of Cdh5 toward the center were observed in localized subregions (Fig. 4E). Quantification of average radial velocity confirmed a significant difference between wild-type and *rasip1* mutants, and distinct pulsatile dynamics in the mutants (Fig. 4F). Further analysis revealed multiple modes of apical Cdh5 behavior in *rasip1* mutants: Cdh5 either dispersed, aggregated within apical compartments, detached from boundaries, or reattached (Fig. S3B and Supplementary Movie 10).

To investigate the fate of apical Cdh5, we selectively photo-converted junctional particles in both wild-type and *rasip1* mutants (Fig. 4G). In wild-type embryos, apical Cdh5 particles dispersed, moved toward the junctional boundaries, and were incorporated into junctional rings (Fig. 4H, arrowheads). This incorporation of Cdh5 clusters, from the apical compartment to junctions, was also observed in *rasip1* mutants (Fig. 4I, arrowhead). However, the incorporation of Cdh5 to junctions in *rasip1* mutants appeared to be more transient compared to wild-type (Fig. 4I(II-IV)). Overall, these findings demonstrate that the loss of Rasip1 leads to increased mobility of Cdh5 from junctions to apical compartments and reduced efficiency in reincorporation back into cell-cell junctions.

### Coordination of contractility in de novo lumen formation

Rasip1 has been proposed to inhibit RhoA or activate Rac1 and Cdc42, thereby either inhibiting or activating actomyosin contractility[14,18,21]. We therefore reasoned that some of the defects in junctional dynamics observed in *rasip1* mutants may be caused by dysregulation of actomyosin contractility. To explore the temporal and spatial activities of actomyosin contractility during de novo lumen formation and distinguish between the diverse roles of Rasip1 in this process, we utilized an NMII reporter, Myl9a-GFP (myosin light chain 9a-GFP) [*Tg(kdrl: Myl9a-GFP)^ip5Tg*][26]. We first performed immunostaining for Myl9a-GFP and Rasip1 at various junctional patches and rings between tip cells (Fig. 5A). Similar to Cdh5, Myl9a strongly colocalized with Rasip1 at early cell-cell contacts but appeared largely segregated in stable junctional rings (Fig. 5A, B). Live imaging of Myl9a and Rasip1 confirmed this pattern, with Rasip1 initially colocalizing with Myl9a at junctional patches (Fig. 5C(I, II), arrows), but later segregating, when Myl9a localized to the periphery and Rasip1 to the center (Fig. 5C(III), arrowheads).

To explore the role of NMII in junctional remodeling, we performed high-resolution live imaging of Myl9a and Cdh5 throughout the patch-to-ring transition during de novo lumen formation. In wild-type embryos, Myl9a primarily localized within the junctional patches before the transition (Fig. 5D(I), arrows, and Supplementary Movie 11). However, Myl9a clusters emerged at the periphery prior to ring formation and were absent from the center (Fig. 5D (II and III), arrowheads). After the nascent junctional rings formed, Myl9a primarily localized along the junctions (Fig. 5D(IV)). Similar transitions in Myosin

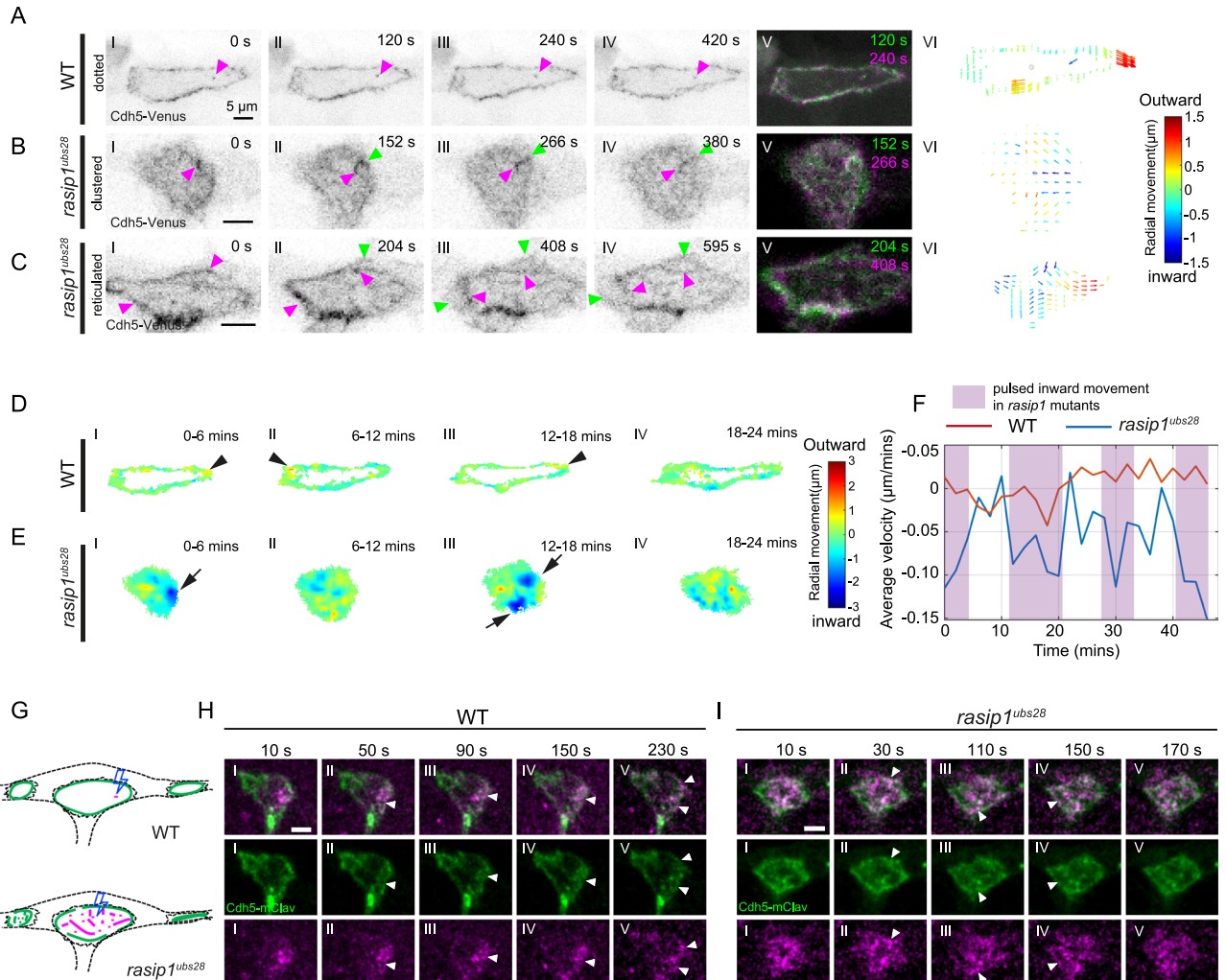

**Fig. 4 | Cdh5 clusters detach from junctions and move toward the apical compartments in *rasip1* mutants. A–C** Time-lapse images of Cdh5-Venus in wild-type embryos (**A**) and in *rasip1^{ubs28}* mutants (**B**, **C**). Magenta arrowheads mark the dots, clusters, or linear structures of Cdh5 within the apical compartments. Green arrowheads mark the newly formed junctions at the boundaries after the detachment of old junctions. (VI) Arrow maps show the speed and direction of the flow of Cdh5 between the two time points in (V). The arrows are twice the length of the actual movement for visualization purposes, with their colors representing velocities relative to the centers of the apical compartments. Consistent observations were made in over 20 samples from 6 independent experiments per group. **D**, **E** Heat maps showing the average velocity of Cdh5 in the radial direction at

6-minute interval in wild-type embryos (**D**) and *rasip1^{ubs28}* mutants (**E**). Average velocity maps were generated from frames with a 36-second interval using PIV and were further combined into a heat map displaying the total particle movement over 360 seconds in the radial direction. Colors in the heat map represent the velocities of movement relative to the centers of the apical compartments. Black arrowheads mark outward movement, while black arrows mark inward movement. **F** Average radial velocity of all Cdh5 in (**D**) and (**E**), measured over 2-minute intervals. **G–I** Diagrams (**G**) and time-lapse images of photo-conversion on the junctional clusters in the apical compartments of wild-type embryos (**H**) and *rasip1* mutants (**I**). White arrowheads mark the converted Cdh5-mClav that moved to the boundaries. All scale bars: 5 μm. Source data are provided as a Source Data file.

localization were confirmed using the Myl9b-Cherry reporter we generated (myosin light chain 9b-mCherry) [*Tg(fli:gal4; UAS: Myl9b-mCherry)^{rk32}*] (Fig. S4). Thus, relocalization of Myl9 appears to precede the formation of a defined junctional ring. In *rasip1* mutants, Myl9a was initially enriched in the junctional patches and subsequently appeared at the periphery, similar to wild-type embryos (Fig. 5E(I, III) and Supplementary Movie 12). However, in *rasip1* mutants, Myl9a soon lost its peripheral localization and reappeared in the junctional patches (Fig. 5E(II, IV), arrows).

We next examined the localization of Myl9a-GFP at junctional rings and reticulated junctions between tip and stalk cells in both wild-type embryos and *rasip1* mutants (Fig. 5F, G). In wild-type embryos, most of the Myl9a-GFP localized along the junctions (Fig. 5F, F′, arrowheads). By contrast, in *rasip1* mutants, Myl9a predominantly localized in the apical domains as clusters (Fig. 5G, G′, arrows).

Quantification of the fluorescence intensities indicated that *rasip1* mutants displayed a higher overall level of Myl9a in the apical compartments compared to wild-type embryos (Fig. 5H). Although the Myl9a signals at the boundaries remained largely the same between wild-type and *rasip1* mutants, their spatial organization was distinct. In wild-type embryos, Myl9a localized linearly along the junctions, whereas in *rasip1* mutants, it formed intermittent clusters along the border (Fig. 5F–H). High-resolution imaging of Myl9a-GFP revealed the dynamics of Myl9a during junctional remodeling. In wild-type embryos, the Myl9a signals primarily accumulated at local hotspots along the junctions, with aggregation of Cdh5, and then dispersed quickly (Fig. 5I, arrowheads, and Supplementary Movie 13). The accumulation of Myl9a at the junctions, along with local straightening or constriction, suggests the involvement of circumferential actomyosin-mediated contractions along junctions, henceforth referred to as

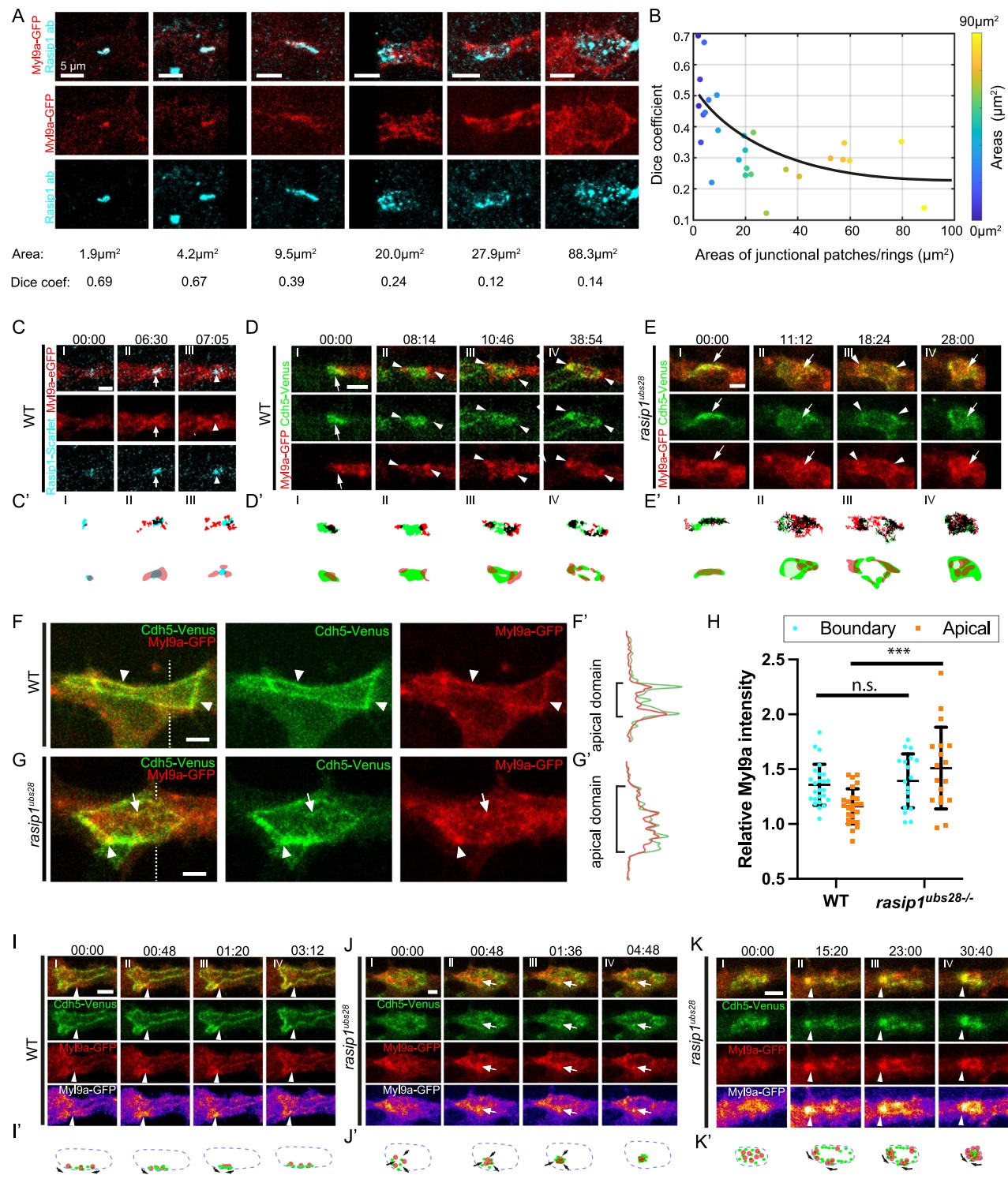

junctional contractility[27]. In contrast, ectopic Cdh5 clusters formed after the local aggregation of Myl9a in the apical compartments (*n* = 7/9 cells in *rasip1* mutants, *n* = 1/12 cells in wild-type embryos) (Fig. 5J and Supplementary Movie 14). Given the significant detachment of Cdh5 clusters in *rasip1* mutants, we hypothesized that radial contractile forces drive the movement of Cdh5 clusters toward the apical compartments, henceforth referred to as apical contractility. In addition to the ectopic apical contractility observed in *rasip1* mutants, we occasionally observed persistent constrictions along junctions turning junctional rings into patches, with sustained enrichment of Myl9a, suggesting a disruption in the organization of contractility at the

boundary regions as well (*n* = 3/10 cells in *rasip1* mutants, *n* = 1/15 cells in wild-type embryos) (Fig. 5K).

## Rasip1 shuttles between apical domains and junctions

The above observations suggest a disrupted organization of contractility within the apical compartment and at junctions in *rasip1* mutants. To further explore the possible roles of Rasip1 in these different cellular compartments, we expressed C-terminal-tagged Rasip1-Scarlet-I, as previously shown in Fig. 1, or N-terminal-tagged GFP-Rasip1 in the zebrafish vasculature (Fig. 6A). The functionality of Rasip1-Scarlet-I was confirmed by its ability to rescue the apical clearance

**Fig. 5 | Rasip1 inhibits contractility at junctional patches and apical compartments. A** Antibody staining for Myl9a-GFP and Rasip1 at junctional patches and rings of varying sizes. Consistent observations in 10 samples from 3 independent experiments. **B** The Dice coefficient and corresponding area of various junctional patches and rings, with data points color-coded by area ($n = 27$ cells from 10 embryos). **C** Time-lapse imaging of Rasip1-scarlet-I and Myl9a-GFP in wild-type embryos during the patch-to-ring transitions. White arrows mark colocalized Myl9a and Rasip1 at the junctional patch, while white arrowheads indicate segregation at later time points. (**C'**) Automatic thresholding and respective graphical representations. Consistent observations in 6 samples from 3 independent experiments. **D, E** Cdh5-Venus and Myl9a-GFP in wild-type embryos (**D**) and *rasip1* mutants (**E**). White arrows indicate Myl9a within the patches, while arrowheads mark Myl9a at the boundary. (**D', E'**) Automatic thresholding and respective graphical representations. Consistent observations in 10 samples from 5 independent experiments per group. **F, G** Distinct localizations of Myl9a-GFP in wild-type embryos (**F**) and *rasip1*

mutants (**G**). Arrowheads label Myl9a at junctions, while arrows indicate Myl9a clusters within the apical compartments. **F', G'** Intensities of signals along the dashed lines in (**F, G**). **H** Normalized Myl9a intensities at the peripheral regions and apical compartments in wild-type embryos and *rasip1* mutants (WT: $n = 25$ cells from 6 embryos; *rasip1*: $n = 18$ cells from 8 embryos), presented as mean ± SD. The Myl9a-GFP intensity is standardized using the mean signal level from the entire cell. ***$P = 0.0001$, n.s., not significant, 2-tailed unpaired t-test. **I–K** Time-lapse images of Cdh5-Venus and Myl9a-GFP showing synchronized clustering in wild-type embryos (**I**) and in *rasip1* mutants (**J, K**). Arrowheads indicate Cdh5 and Myl9a along the junctions, while arrows indicate the Cdh5 and Myl9a clusters within the apical compartment. (**I'**) Diagrams illustrate transient Cdh5 aggregation under Myl9a enrichment along the junctions in wild-type embryos. (**J', K'**) Diagrams illustrate Cdh5 aggregation under Myl9a enrichment towards the apical compartments (**J'**) or persistent contraction along junctions in *rasip1* mutants (**K'**). All scale bars: 5 μm. Source data are provided as a Source Data file.

defects (Fig. S5A, S5B). Consistent with antibody staining, Rasip1-Scarlet-I was highly confined within the apical compartments (Fig. 6B, C). However, Rasip1 distribution within the apical membrane was not uniform, showing clustered or linear arrangements proximal to junctions, as evidenced by antibody staining and Rasip1 reporter analyses (Fig. 6B, C, and S5C). Live imaging of Rasip1 reporters revealed that Rasip1 was highly enriched along the local regions of junctions undergoing constriction (Fig. 6D(I-III), arrowheads, and Supplementary Movie 15). Conversely, we also observed strong enrichment of Rasip1 as clusters inside the apical compartments during constriction of the entire apical compartment (Fig. 6E(I-III), arrows, and Supplementary Movie 15). Before and after local or global constrictions, Rasip1 exhibited dynamic relocalization between the apical compartments and junctions. Rasip1 translocated from the apical compartment to the local boundary during constriction, with the latter's contour outlined by Rasip1 itself and mRuby2-UCHD, an F-actin marker (Fig. 6F(II-IV), arrowheads, and Supplementary Movie 16)[28]. After local contractions, junctional Rasip1 translocated back to the apical domains (Fig. 6D(III, IV), 6 G and Supplementary Movie 17). These observations suggest that Rasip1 shuttles between apical and junctional compartments. One possibility for such dynamic movement is the local clustering of the apical membrane. To test this hypothesis, we performed live imaging of Rasip1 alongside the apical marker Podxl1. Our data showed that Podxl1 was not dynamically shuttled from the apical compartments to junctions, suggesting that the dynamic translocation of Rasip1 is not driven by local apical membrane aggregation (Fig. 6H, arrowheads).

Similar to the segregation observed in junctional patches, Rasip1 first colocalized and later segregated with Myl9a within the apical compartments (Fig. 6I and Supplementary Movie 18) and along junctions (Fig. 6J and Supplementary Movie 19). The clustering of Rasip1 and Myl9a was highly dynamic, with new clusters forming while old clusters disappeared rapidly (Fig. 6I, J). We observed dynamic recruitment of Rasip1 to newly formed Myl9a clusters (Fig. 6I(II, III)). Both Rasip1 and Myl9a clusters diminished after their coexistence at Myl9a hotspots (Fig. 6I(IV), J(IV)). Thus, Rasip1 appears to dynamically respond to local high contractility at junctions and apical compartments. In combination with functional studies of Rasip1, we hypothesize that Rasip1 inhibits contraction by localizing to contracting sites either at the apical compartments or at junctions. The presence of Rasip1 does not immediately halt contraction (Fig. 6D, F). Instead, it seems to play a critical role in moderating the activity of the contractility machinery. Supporting this hypothesis, we observed sustained Myl9a and actin clustering at junctions and within apical compartments in *rasip1* mutants (Fig. 5J, K, and S6A-D).

## Modulating contractility alters apical clearance defects

The above observations indicate a local ectopic elevation of actomyosin contractility within the apical compartments in *rasip1* mutants.

To explore whether the mislocalization of Cdh5 in *rasip1* mutants is caused by increased actomyosin tension, we first reduced actomyosin tension through acute pharmacological inhibition of ROCK using Y-27632. Strikingly, ROCK inhibition was sufficient to rescue several aspects of the *rasip1* mutant phenotypes. The initial adhesion sites opened properly and maintained clear ring shapes 80 minutes post-treatment, similar to control embryos (Fig. 7A–C). The reticulated junctions that had formed prior to drug treatment were also eliminated, and the previously fuzzy cell-cell junctions became more defined (Fig. 7D, E, and S7). Furthermore, the apical compartments were cleared of junctional proteins, and the elevated apical Myl9a was reduced under acute inhibition of ROCK (Fig. 7D).

Conversely, we tested whether the apical clearance defects in *rasip1* mutants can be phenocopied by elevating cortical actomyosin contractility specifically in the apical compartment. To this end, we overexpressed the catalytic domain of ROCK1 (N-ROCK1) at the apical membrane by tagging it to the apically localized protein Podxl1. Myl9a-GFP clustering was observed alongside N-ROCK1-Scarlet-Podxl1 at the apical compartments, confirming increased contractility in response to ROCK activation (Fig. S8A). In agreement with this prediction, Cdh5 ectopically localized within the apical compartments upon the expression of N-ROCK1-Scarlet-Podxl1 (Fig. 7F, G). Unlike the evenly distributed Scarlet-Podxl1 observed in the apical membrane, N-ROCK1-Scarlet-Podxl1 exhibited heterogeneous aggregations reminiscent of Podxl1 distribution in *rasip1* mutants (Fig. 7F, F', F").

To achieve precise temporospatial control of RhoA/ROCK activities, we applied a single-component optogenetic tool for inducible RhoA GTPase signaling in the zebrafish vasculature, *Tg(fli1a: RhoA-BcLOV4-mCherry)*[29]. Local RhoA signaling was activated under a 450 nm laser at hand-drawn ROIs. Following continuous 450 nm laser activation in wild-type embryos, Myl9a-GFP clustering was observed alongside Opto-RhoA-mCherry, confirming increased contractility in response to RhoA activation (Fig. S8B). Similarly, laser activation at the apical compartments induced the formation of Cdh5 clusters together with RhoA-BcLOV4-mCherry, while Cdh5 at the boundaries became fractured and diffuse (Fig. 7H, I, and Supplementary Movie 20). In summary, these findings suggest that enhanced apical contractility leads to apical clearance defects. In combination with the observation of ectopic apical contractility and collective Cdh5 detachment in *rasip1* mutants, we propose that Rasip1 stabilizes cell-cell junctions by confining Cdh5 clusters to junctions through inhibiting apical contractility.

## Heg1 and Krit1 regulate the dynamic recruitment of Rasip1

We have identified Rasip1 as one of the earliest apical proteins recruited to the initial cell-cell contact site, where it assists in segregating junctions from the apical membranes by inhibiting actomyosin contractility at the center. After the establishment of the apical domain, Rasip1 dynamically relocates between the apical compartments and the junctions to regulate local hypercontractility. One key

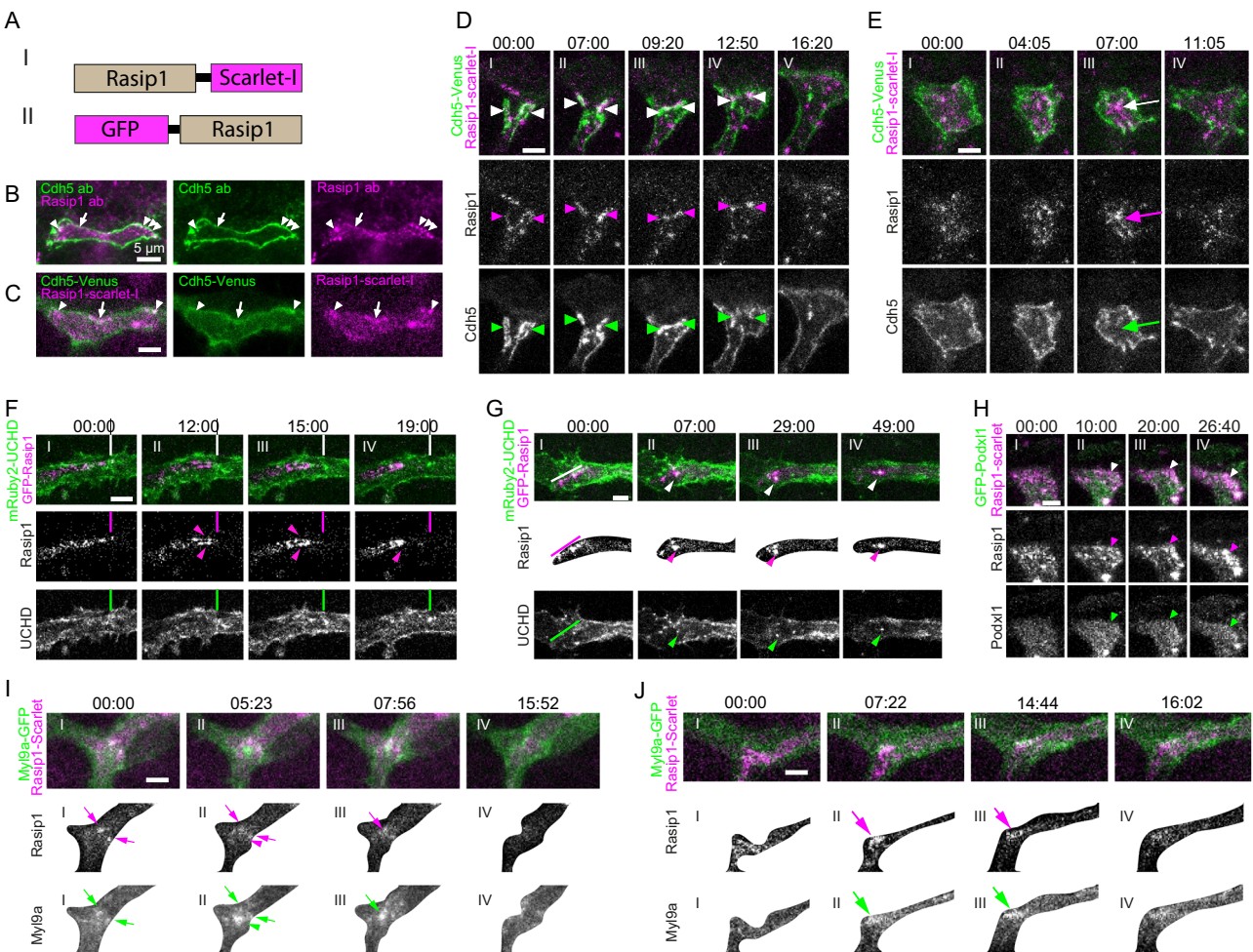

**Fig. 6 | Rasip1 shuttles between junctions and apical compartments in close association with Myosin. A** Schematic drawing of recombined Rasip1-Scarlet-I and GFP-Rasip1. **B** Antibody staining for Rasip1 and Cdh5 in wild-type embryos at 32 hpf. Consistent observations were made across 14 samples from 4 independent experiments. **C** Expression of Rasip1-Scarlet-I and Cdh5-Venus in wild-type embryos at 32 hpf. In both (**B**) and (**C**), white arrowheads indicate Rasip1 clusters, while white arrows label linear Rasip1 localizations along the junctions. Consistent observations were made across 22 samples from 7 independent experiments. **D** Time-lapse images of Cdh5-Venus and Rasip1-Scarlet-I with Rasip1 enriched at constricting junctions (arrowheads). **E** Time-lapse images of Cdh5-Venus and Rasip1-Scarlet-I with Rasip1 enriched at the apical compartment (arrows). **F** GFP-Rasip1 relocating from the apical compartment to the constricting junctions. Lines label the position of the apical compartment before contraction. Arrowheads mark the GFP-Rasip1

enrichment at the constricting junctions. **G** GFP-Rasip1 relocating from junctions to apical compartments as clusters. Lines indicate GFP-Rasip1 localized near the anterior boundary of the apical compartment. Arrowheads mark the GFP-Rasip1 clusters within the apical compartment. Masks were applied to focus solely on the apical domain. Consistent observations were made in 15 samples from 7 independent experiments (**D**–**G**). **H** Time-lapse images of GFP-Podxl1 and Rasip1-Scarlet-I during constrictions. Arrowheads indicate highly enriched Rasip1 at the constricting sites. **I**, **J** Time-lapse images showing the local enrichment of Myl9a-GFP and Rasip1-Scarlet-I and subsequent depletion at apical compartments (**I**) or along junctions (**J**). Arrows mark clusters where Rasip1 and Myl9a overlap, while arrowheads indicate Myl9a-positive, Rasip1-negative clusters. Masks were applied to focus solely on the apical domain. Consistent observations were made in 8 samples from 3 independent experiments. All scale bars: 5 μm.

unanswered question is how Rasip1 achieves such specific and dynamic localization. Rasip1 and Krit1 (CCM1), another key regulator of Rho GTPases, bind to different sites on the cytoplasmic domain of Heg1, a transmembrane protein (Fig. 8A)[30,31]. The interactions between Heg1 and Rasip1 have been proposed to facilitate recruitment of Rasip1, leading to the stabilization of endothelial cell-cell junctions[31]. It has also been reported that Krit1 is recruited to cell-cell junctions and forms complexes with Cdh5[32–35].

Our previous study demonstrated that the loss of *heg1* or *krit1* leads to zig-zag or contorted cell-cell junctions, in sharp contrast to *rasip1* mutants[27]. In contrast, junctional rings appeared more circular in *cdh5* null mutants[36]. Despite these distinct shapes of cell-cell junctions, the segregation of apical compartments from junctions remained largely unaffected in *heg1*, *krit1*, and *cdh5* mutants[27,37]. This allowed us to explore the dynamic recruitment of Rasip1 in these loss-of-function mutants through Rasip1 immunostaining and live

imaging[36,38,39] (Fig. 8B–E). Interestingly, the recruitment of Rasip1 to the initial cell-cell contact sites and apical compartments appeared largely independent of Heg1, Krit1, and Cdh5 (Fig. 8B). However, Rasip1 was poorly localized at junctions in *heg1* and *krit1* mutants, as shown by antibody staining and live imaging, suggesting that Heg1 and Krit1 are involved in the dynamic translocation of Rasip1 between the junctional and apical compartments (Fig. 8C–E; Supplementary Movie 21). Notably, Rasip1 was strongly enriched at junctions in *cdh5* mutants, suggesting the involvement of other junctional proteins in positioning Krit1-Heg1-Rasip1 complex to junctions (Fig. 8C, D).

### Rasip1 translocates in respond to increased contractility
Rasip1 is first localized at the initial cell-cell adhesion sites together with Myl9a (Fig. 5A–C). Following the formation of stable junctional rings, Rasip1 dynamically translocates to newly formed Myl9a clusters and constricting junctions (Fig. 6F, I(II and III)). These observations

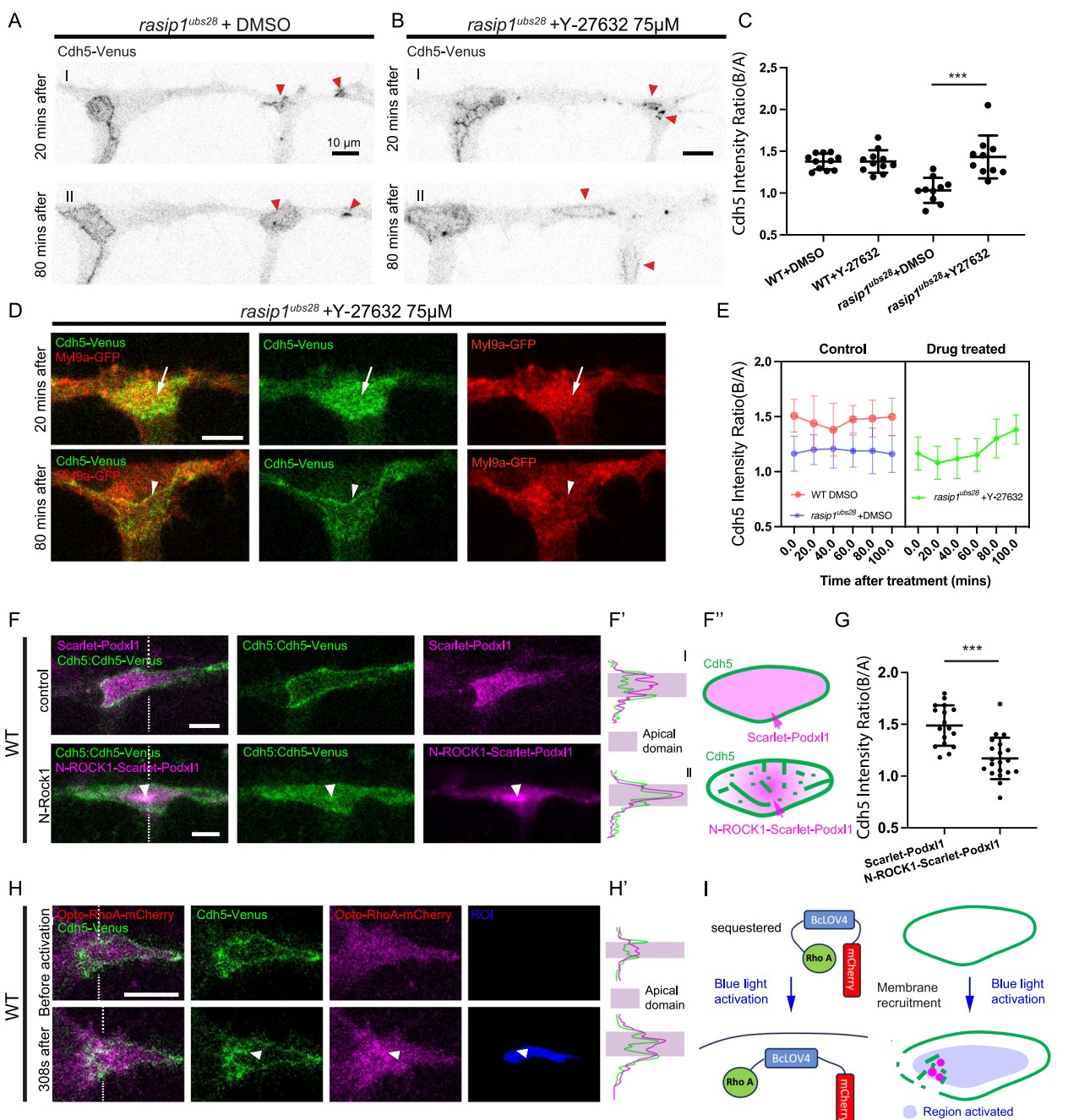

**Fig. 7 | Precise control of apical contractility is required for apical clearance.** Acute DMSO (**A**) or Y-27632 (**B**) treatment on *rasip1* mutants with unopened cell-cell contact sites. Red arrowheads indicate the same junctional patches/rings at 20 minutes and 80 minutes after drug treatment. **C** The Cdh5 boundary-to-apical ratio in the following groups: WT + DMSO (*n* = 11 cells from 5 embryos), WT + Y-27632 (*n* = 11 cells from 5 embryos), *rasip1* + DMSO (*n* = 10 cells from 5 embryos), *rasip1* + Y-27632 (*n* = 10 cells from 6 embryos) at 200 minutes after treatment for newly opened junctions. Data are presented as mean ± SD. \*\*\**P* = 0.0005, 2-tailed unpaired t-test. **D** Myl9a-GFP and Cdh5-Venus at 20 minutes and 80 minutes after acute Y-27632 treatment (75 μM) in *rasip1^ubs28^* mutants. White arrows label ectopic apical Myl9a and Cdh5, while white arrowheads mark junctional Myl9a at 80 minutes post-treatment. Consistent observations were made across 8 samples from 3 independent experimental setups. **E** The Cdh5 boundary-to-apical ratio over the acute treatments in the following groups: WT + DMSO (*n* = 16 cells from 6

embryos), *rasip1* + DMSO (*n* = 20 cells from 6 embryos), *rasip1* + Y-27632 (*n* = 10 cells from 4 embryos). Data are presented as mean ± SD. **F** Wild-type embryos expressing Scarlet-I-Podxl1 or N-ROCK1-Scarlet-I-Podxl1 (**F**) and corresponding diagram (**F''**). (**F'**) Intensities of Cdh5-Venus and Scarlet-I-Podxl1 or N-ROCK1-Scarlet-I-Podxl1 along the dashed lines in (**F**). White arrowheads label enriched N-ROCK1-Scarlet-I-Podxl1 and Cdh5 within the apical domain. **G** Quantification of the Cdh5 boundary-to-apical ratio in wild-type embryos expressing Scarlet-I-Podxl1 (*n* = 17 cells from 9 embryos) or N-ROCK1-Scarlet-I-Podxl1 (*n* = 20 cells from 8 embryos). Data are presented as mean ± SD. \*\*\**P* < 0.0001, 2-tailed unpaired t-test. **H, I** Activation of Opto-RhoA selectively at the apical compartment (**H**) and corresponding diagram (**I**). White arrowheads label apical Cdh5 and Opto-RhoA clusters. (**H'**) Intensities of Cdh5-Venus and Opto-RhoA-mCherry along the dashed lines in (**H**). All scale bars: 10 μm. Source data are provided as a Source Data file.

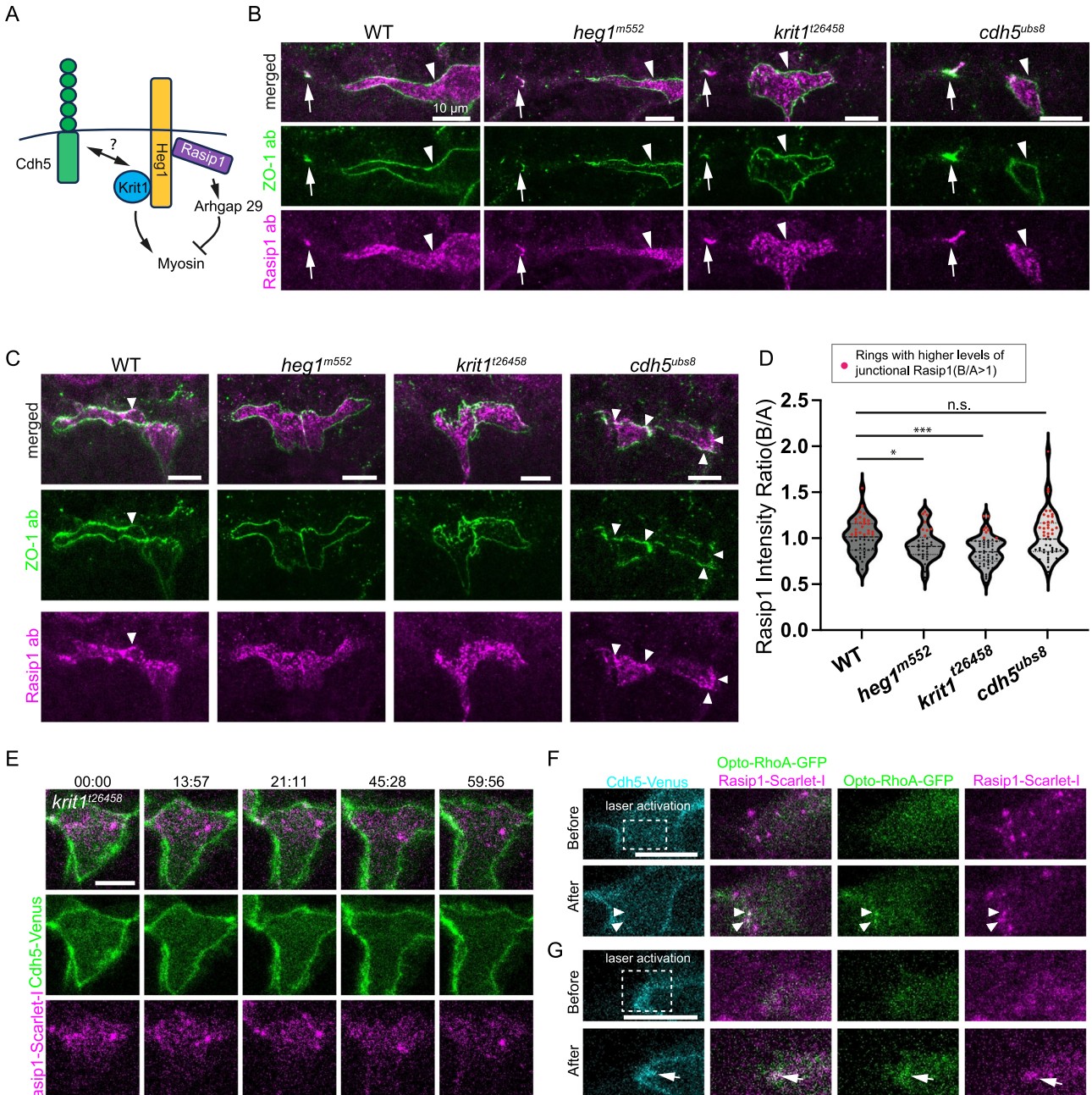

**Fig. 8 | Dynamic recruitment of Rasip1 through Heg1 and Krit1 and its response to contractility. A** Diagram showing the interactions between Rasip1, Heg1, Krit1, and Cdh5. **B** ZO-1 and Rasip1 antibody staining in wild-type embryos, *heg1^m552^, krit1^t26458^* and *cdh5^ubs8^* mutants at 30 hpf. White arrows label the initial cell-cell contact sites between tip cells with enriched Rasip1. White arrowheads mark the apical compartments with enriched Rasip1. **C** ZO-1 and Rasip1 antibody staining in wild-type embryos, *heg1^mSS2^, krit1^t26458^* and *cdh5^ubs8^* mutants at 32 hpf. White arrowheads indicate Rasip1 enrichment along the junctions. **D** Violin plot showing the Rasip1 boundary-to-apical ratio of average intensity per ring in wild-type embryos (*n* = 49 cells from 6 embryos), *heg1^mSS2^*(*n* = 33 cells from 7 embryos), *krit1^t26458^* (*n* = 57 cells from 7 embryos), and *cdh5^ubs8^* mutants (*n* = 47 cells from 8 embryos). Data are presented as mean ± SD. \*\*\**P* = 0.0002, \**P* = 0.0492, n.s., not significant, 2-tailed unpaired t-test. **E** Time-lapse images of Cdh5-Venus and Rasip1-Scarlet-I in *krit1* mutants showing poor enrichment of Rasip1-scarlet-I along the junctions. **F, G** Activation of Opto-RhoA induced Rasip1-Scarlet-I enrichment at the apical compartment (**F**) and junctions (**G**). White arrowheads and arrows indicate the colocalized Rasip1-Scarlet-I and Opto-RhoA-Cherry clusters at the apical compartment and junctions, respectively. All scale bars: 10 µm. Source data are provided as a Source Data file.

suggest that Rasip1 is dynamically recruited, potentially driven by increased local contractility. To further explore this hypothesis, we used Opto-RhoA-GFP to selectively activate RhoA in the apical compartments or at junctions (Fig. 8F, G). We observed the dynamic recruitment of Rasip1-Scarlet-I, which colocalized with RhoA-BcLOV4-GFP clusters within the apical compartments (Fig. 8F, arrowheads, and S9A) and junctions after activation (Fig. 8G, arrows, and S9B) (*n* = 4 experiments).

experiments). These findings suggest that Rasip1 is dynamically recruited in response to increased contractility.

## Discussion

The removal of preapical adhesions is a crucial step in apical compartment formation during de novo lumen formation. VE-cadherin, the primary endothelial cell-cell adhesion molecule, responds to

mechanical forces generated by the actomyosin cytoskeleton[27,40–42]. A previous study on vasculogenesis in mouse embryos suggested that junctional complexes are pulled from the preapical patch to the periphery by actomyosin contractility[14]. Consistent with this, we observed Cdh5 redistribution from the preapical patch to the periphery through photoconversion of junctional patches. However, we found that in addition to redistribution, proper restriction of junctional clusters is necessary for the persistent clearance of junctional patches, which is mediated by precise control of both junctional and apical contractility. In wild-type embryos, most of the Myl9a and Myl9b proteins localize along the junctions, driving the circumferential movement of Cdh5 along junctional rings. In contrast, *rasip1* mutants exhibited disorganized apical and junctional contractility. Although the level of boundary-localized Myl9a in *rasip1* mutants remained comparable to that in wild-type embryos, Myl9a exhibited distinct clustered localization and persistence at junctions, leading to a collapse of a portion of junctional rings. Additionally, apical contractility is elevated in *rasip1* mutants and the ectopic apical actomyosin activity correlates with the detachment of junctional clusters from the boundary into the apical compartments. This detachment of Cdh5 exhibits pulsatile behavior and leads to apical clearance defects. By carefully manipulating contractility, these apical clearance defects in *rasip1* mutants can either be rescued or phenocopied; pharmacological inhibition of ROCK successfully rescued the apical clearance defects in *rasip1* mutants, whereas activation of RhoA or ROCK in the apical compartments induced similar defects to those seen in *rasip1* mutants. Overall, our studies reveal that Rasip1 regulates the proper segregation of proteins within the apical compartments and junctions by precisely coordinating apical and junctional contractility.

## Generating clean apical surfaces by Rasip1 for lumen formation

The process of lumen formation during vasculogenesis and angiogenesis involves distinct stages[2]. Initially, endothelial cells establish cell-cell adhesion, followed by junctional relocalization, leading to the development of apical interfaces between adjacent endothelial cells flanked by junctions. Subsequently, the incipient apical surfaces between two endothelial cells repel from each other, creating a small slit that initiates the formation of luminal space. Our investigation highlights the involvement of Rasip1 in precisely regulating junctional and apical contractility to facilitate the partitioning of junctions and apical domains. On the other hand, the proper separation of luminal surfaces relies on Podxl1 and CD34, which carry negatively charged sialic acid[2,43]. Neutralizing this sialic acid charge demonstrated the role of Podxl1 in electrostatic repulsion during vascular lumen formation[44]. In our study, we found that Rasip1 is initially recruited to junctional patches before the recruitment of Podxl1. This observation aligns with the distinct molecular interactions of the two proteins with the cytoskeletal network. Podxl1 associates with ERM proteins (Ezrin, Radixin, and Moesin), which link this transmembrane protein to the actin cytoskeleton, whereas Rasip1 is dynamically recruited to junctions via Heg1 and potentially other interacting partners[2,31,45,46]. Rasip1 facilitates the proper segregation of the apical membrane from junctions, thus enabling efficient Podxl1 recruitment to the apical domain, which is crucial for repelling apical surfaces. In the absence of Rasip1, Podxl1 exhibits aberrant local aggregations at the presumed apical compartments, possibly due to enhanced apical contractility, and becomes inadequately restricted to the apical domain.

## Defining cellular compartments by differential contractility

One of the fundamental questions in cell biology is how distinct cellular compartments are established and maintained to achieve precise spatiotemporal regulation of numerous processes and pathways. Convincing evidence has emerged from in vitro measurements on reconstructed model membranes showing that actomyosin contraction mediates the macroscopic de-mixing of membrane components

and membrane compartmentalization[47–49]. However, it remains largely unknown whether and how actomyosin contractility alters the segregation behavior of different cellular domains in vivo. In this study, we identified the roles of Rasip1 in the proper partitioning of the apical and junctional compartments by defining different levels of actomyosin contractility. To our knowledge, Rasip1 is one of the earliest apical proteins recruited to the initial cell-cell contact sites (Fig. 9A and S10). The recruitment of Rasip1 to the initial cell-cell contact site and the apical domain is largely independent of Heg1, Krit1, and Cdh5. In contrast, our data suggest that Rasip1 dynamically responds to high contractility and acts to suppress it. Rasip1 and Myosin rearrange dynamically within the junctional patches and begin to segregate from each other (Fig. 9A). The centrally localized Rasip1 inhibits Myosin, while Myosin localized at the periphery pulls the junctional complexes toward the borders (Fig. 9A and S10). After the establishment of apical compartments, Rasip1 further prevents the detachment of junctional clusters and suppresses persistent constrictions by inhibiting apical and junctional contractility, thus stabilizing cell-cell junctions (Fig. 9A). In contrast, in the absence of Rasip1, the junctional clusters are not consistently relocated to the periphery under increased apical contractility in the pre-apical patches (Fig. 9B). Instead, ectopic contractility along radial directions leads to the abnormal aggregation of junctional clusters at the apical compartments and boundary regions in *rasip1* mutants (Fig. 9B).

## Tuning contractility with the same pool of Rasip1 at distinct compartments

Canoe, a homologous protein of Rasip1 in *Drosophila*, displays a multifaceted spatial organization involving membrane localization, recruitment to nascent junctions, and macromolecular assembly at tricellular junctions, and is implicated in regulating E-cadherin redistribution under the control of Rap1[50,51]. The distinct localizations of Canoe were proposed to respond to different levels of external forces, thus stabilizing cell-cell junctions or facilitating cell-cell rearrangements[52]. Similarly, Rasip1 acts as a Rap1 effector, regulating the integrity of new vessels and endothelial barrier functions through ArhGAP29, a GTPase-activating protein for Rho proteins[16,17,21]. The proper function of Rasip1 is mediated by its recruitment to junctions, thereby stabilizing cell-cell junctions through direct binding with Heg1[31]. In this study, we identified dynamic relocations of Rasip1 between the apical compartments and local junctions during de novo lumen formation. The shuttling of Rasip1 regulates the organization of contractility in both compartments and the redistribution of Cdh5. Like Canoe, Rasip1 dynamically responds to high contractility at distinct cellular compartments through rapid recruitment. The shuttling of Rasip1 between the apical compartments and junctions might convey another layer of dynamic regulation at the compartment level. The dominant apical localization and the rapid responsiveness of Rasip1 at junctions define and orchestrate different levels of actomyosin contractility in distinct compartments in a robust manner, utilizing the same pool of Rasip1. This occurs without the necessity of balancing two regulators at separate compartments. However, how Rasip1 and Canoe sense high tension/contractility remains largely unknown. Previous studies have identified a key role for Heg1 and Krit1 in maintaining junctional contractility through their dynamic recruitment to local regions of junctions and promoting ROCK2 interactions with cell-cell junctions[27,53]. In the absence of either of these two proteins, the dynamic recruitment of Rasip1 to junctions is disrupted, indicating that Heg1 and Krit1 play crucial roles in Rasip1 recruitment to junctions (Fig. 9A). The interactions between Rasip1, Heg1 and Krit1, as well as the roles of Heg1 and Krit1 in maintaining junctional contractility, suggest the possibility that Rasip1 responds to high contractility at junctions by engaging actomyosin activators through Heg1 and Krit1. However, how Rasip1 is properly localized to the initial cell-cell contact sites and the apical compartments remains largely

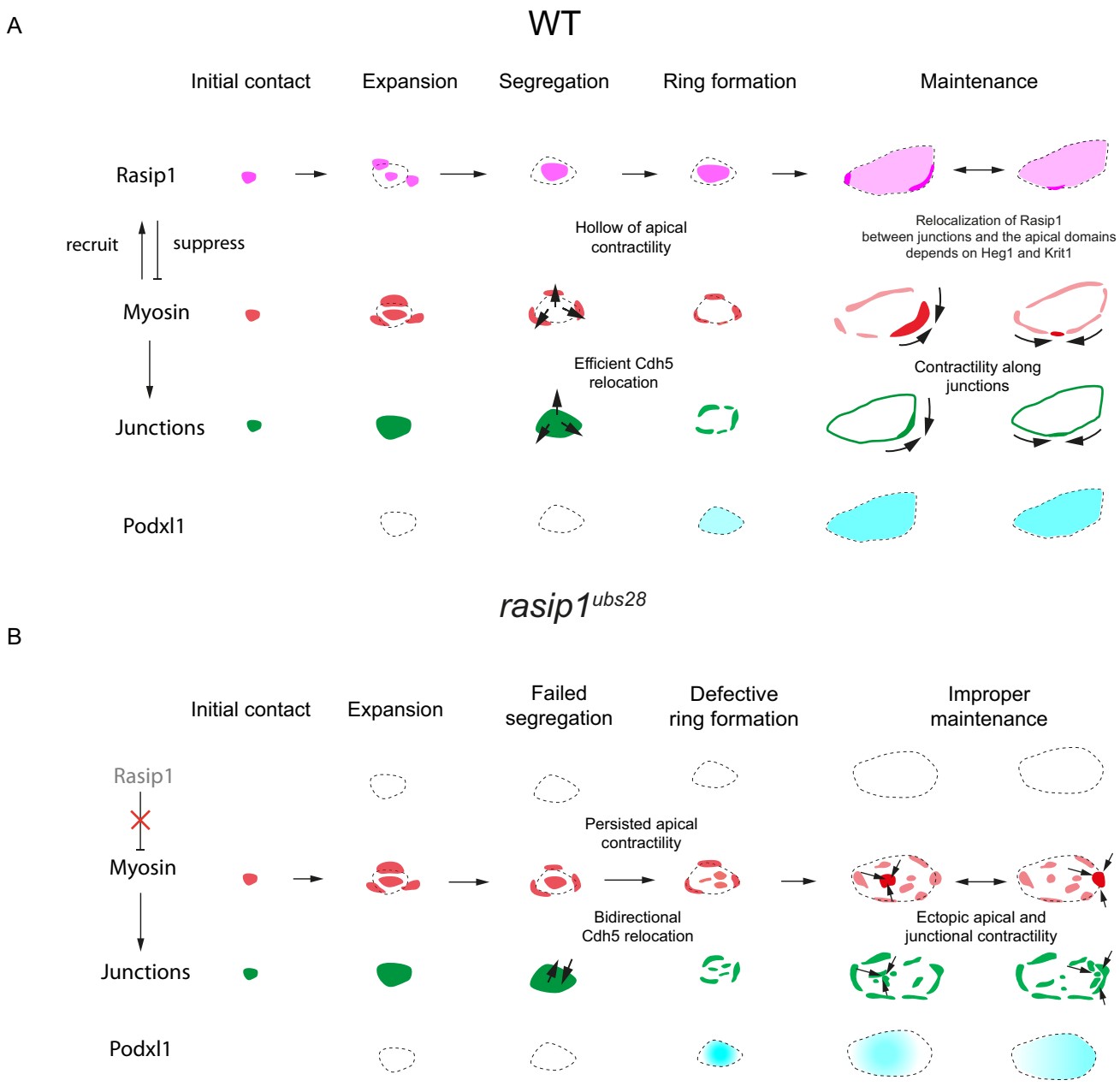

**Fig. 9 | Initiation and maintenance of apical compartments via differential contractility regulated by the dynamic recruitment of Rasip1. A** Diagrams showing the molecular steps of de novo lumen formation. In wild-type embryos, both Myosin and its inhibitor Rasip1 are recruited to the initial contact sites (initial contact). The initial contact sites expand into junctional patches (expansion). Rasip1 induces differential contractility at the junctional patches by inhibiting NMII at the center (segregation). Thus, Cdh5 effectively relocates toward the periphery, pulled by surrounding Myosin, generating the clear apical domain (ring formation). Once the apical compartments are established, Rasip1 shuttles between junctions and apical compartments in response to local high tension, tuning contractility at both compartments. Rasip1 confines Cdh5 to junctions by suppressing apical contractility, maintaining the clear segregation of apical compartments from junctions. Conversely, the recruitment of Rasip1 to junctions, controlled by Heg1 and Krit1, is critical for moderating contractility along junctions. **B** In *rasip1* mutants, the pre-apical Myosin fails to be inhibited, leading to ectopic radial contractility toward the center and unsustainable Cdh5 relocation toward the periphery (failed segregation and defective ring formation). The ectopic radial contractility in *rasip1* mutants de-stabilizes junctional proteins at boundaries and induces ectopic aggregations of junctional clusters at both apical compartments and boundary regions (improper maintenance).

unknown. Further studies are required to identify other linker proteins in different cellular compartments.

## Methods
### Fish maintenance and stocks
Zebrafish (*Danio rerio*) were maintained according to FELASA guidelines[54]. All experiments were performed in accordance with federal guidelines and were approved by the Kantonales Veterinäramt of Kanton Basel-Stadt (1027H, 1014HE2, 1014 G). The following new zebrafish lines were developed in this study: *Tg(cdh5: cdh5-mClavGR2)ubsS8*, *Tg(fli:gal4; UAS: Myl9b-mCherry)rk32*, *Tg(fli1a: Rasip1-Scarlet-I)ubs59*. The previously established zebrafish lines used were: *Tg(cdh5: cdh5-TFP-TENS-Venus)uq11bh22*, *TgKl(tjp1a-tdTomato)pd12423*, *Tg(fli1a: GFP-Podxl1)ncu530Tg24*, *Tg(kdrl: Myl9a-GFP)ip5Tg26*, *Tg(fli1ep:gal4ff)ubs355*, *Tg(UAS:mRuby2-UCHD)ubs2028*, *rasip1ubs2820*, *heg1mS5238*, *ccm1t264S8 39* and *cdh5ubs8* mutants[36]. Transient expressions in zebrafish embryos

included: *fli1a: GFP-Rasip1*, *fli1a: RhoA-BcLOV4-mCherry*, *fli1a: RhoA-BcLOV4-eGFP*. Sex was not considered in this study because zebrafish embryos used for the experiments were at developmental stages prior to sex differentiation. Therefore, the sex of the embryos was not determined or relevant to the outcomes.

## Cloning and transgenesis of Tg(cdh5: cdh5-mClavGR2)[ubs58]

The sequence encoding *mClavGR2* was introduced into exon 13 of *cdh5* using BAC recombineering technology, positioned between the p120-catenin and β-catenin binding sites of the translated Cdh5 protein. This position has been shown to be harmless to the proper function of Cdh5[22]. A defective λ prophage system was employed for BAC recombineering[56]. The BAC CH73-357K2 containing the *cdh5* gene was transformed into SW102 *E. coli*. SW102 cells containing the BAC were then inoculated and transformed with the *galK* gene flanked by homology arms. One surviving colony was subsequently inoculated and transformed with the insert containing *mClavGR2*, which was flanked by the same homology arms as before. Transformed cells were selected on DOG-containing minimal plates (negative selection for *galK* expression). Surviving colonies were analyzed for the desired modification. The selected cdh5-mClav-containing BAC was subsequently tested by XbaI digestion, followed by PCR analysis and sequencing. The primers used for this study were as follows: Galk_mClav_fo (TGGCCATGATGATCGAGGTGAAGAAGGACGAGGCAGATCGTGATCGAGATCCTGTTGACAATTAATCATCGGCA), Galk_mClav_re (GATTCGGTTCCCTCATAGCCATAGATGTGTAATGTGTCATAGGGAATGCCTCAGCACTGTCCTGCTCCTT), Homo_mClav_fo (TGGCCATGATGATCGAGGTGAAGAAGGACGAGGCAGATCGTGATCGAGATGCCGGCGTGAGCAAGGGCGAGGAG), and Homo_mClav_re (GATTCGGTTCCCTCATAGCCATAGATGTGTAATGTGTCATAGGGAATGCCGCCGGCCTTGTACAGCTCGTCCATGCC).

## Cloning and transgenesis of Tg(fli1a:Rasip1-scarlet-I)[ubs59]

The cDNA of Rasip1 was cloned from a zebrafish cDNA library into the pME vector through the BP reaction of gateway cloning using the following primers: rasip1-BP-F (GGGGACAAGTTTGTACAAAAAAGCAGGCTCTCGAGatggaggagtctggcagtc) and rasip1-BP-R (GGGGACCACTTTGTACAAGAAAGCTGGGTttacagtctcgtccgct). Restriction sites for NotI and MfeI were introduced into *pME-Rasip1* through PCR at the 3' end of the Rasip1 cDNA using the following primers: pME-Rasip1-3R-NotI-F (AATCATGCGCGGCCGCTAAAACCCAGCTTTCTTGTACAAAGTTGGCATTATAAG) and pME-Rasip1-3R-MfeI-R (ATACATACCAATTGGCCTCCGGATCCGCCCAGTCTCGTCCGCTGATTAG). The cDNA of Scarlet-I was synthesized from Genewiz (Azenta, USA) with codon optimization for zebrafish. Restriction-ligation was employed for the insertion of Scarlet-I at the C-terminus of Rasip1 with the linker GGSGGQL using the following primers: Scarlet-I-F (ATACGTGGCAATTGATGGTGAGCAAGGGCGAG) and Scarlet-I-R (TTATCCAAGCGGCCGCCGCTTCCTCCTCCGCTTCCTCCcttgtacagctcgtccat). The constructed *pME-Rasip1-Scarlet-I* was fully sequenced. Subsequently, *pME-Rasip1-Scarlet-I* was cloned into *PDestTol2CG2* through the LR reaction of gateway cloning, incorporating a 2 kb *fli1a* promoter and a 3' poly(A) tail.

## Cloning of fli1a:GFP-Rasip1

*eGFP* was inserted at the 5' end of *Rasip1* in the *pME-Rasip1* vector using the linker GGSGGGSGG through Gibson assembly with the following primers: GFP-Gib-F (GTACAAAAAAGCAGGCTCATGGTGAGCAAGGGCGAG), GFP-Gib-R (ctgccagactcctccatCCCTCCGCTTCCTCCTCCGCTTCCTCCCCTTGTACAGCTCGTCCATGC), pME-rasip1-Gib-F (GatggaggagtctggcagtcC), and pME-rasip1-Gib-R (GCCTGCTTTTTTTGTACAAAGTTGGCAT). The constructed *pME-GFP-Rasip1* was fully sequenced. Subsequently, *pME-GFP-Rasip1* was cloned into *PDestTol2CG2* through the LR reaction of gateway cloning, incorporating a 2 kb *fli1a* promoter and a 3' poly(A) tail.

## Cloning of fli1a: RhoA-BcLOV4-mCherry

The light-oxygen-voltage (LOV) flavoprotein BcLOV4, derived from *Botrytis cinerea*, swiftly relocates to the plasma membrane through an electrostatic interaction with lipids on the inner leaflet, a process regulated by blue light[57,58]. It was used as a single-component optogenetic tool for inducible RhoA GTPase signaling in cell culture[29]. We ordered the *opto-RhoA-mCherry_pcDNA3.1* construct from Addgene (#164472). The Tol2 vector *PDestTol2CG2*, which carries a 2 kb *fli1a* promoter, was amplified with PCR to incorporate the restriction sites EcoRI and NheI. *RhoA-BcLOV4-mCherry* was cloned into the Tol2 destination vector *PDestTol2CG2* by adding the restriction sites MfeI and XbaI through PCR using the following primers: tol2-NheI-F (GATTTTCTGCTAGCGgtaccatcgatgatgatccag), tol2-fli-EcoRI-R (TTAGTACTGAATTCAATTCCACCGCGTCTGAATTAATTCCAGCC), opto-RhoA-mfeI-F (GTCAGTCACAATTGatgtcagctgccatccgg), and opto-RhoA-xbaI-R (ACTGACTGTCTAGAttacttgtacagctcgtccatgc).

## Cloning of fli1a: RhoA-BcLOV4-eGFP

The *mCherry* in the original *opto-RhoA-mCherry_pcDNA3.1* construct was replaced with *eGFP* using Gibson assembly with the following primers: GFP-gib-F (ATGGTGAGCAAGGGCGAGG), GFP-gib-R (cttgtacagctcgtccatgcCG), mCherry-3-F (ggcatggacgagctgtacaagtaa), and mCherry_5_R (tcctcgcccttgctcaccat). *RhoA-BcLOV4-eGFP* was cloned into the Tol2 vector under a 2 kb *fli1a* promoter using the same strategy described above.

## Live imaging, photo-conversion and optogenetics

Zebrafish embryos between 30 to 32 hpf were dechorionated and anaesthetized with 0.16 mg ml⁻¹ (1×) tricaine methanesulfonate (Sigma). The embryos were mounted into microwell dishes within 0.75% low-melting-point agarose (ROTI) and covered with E3 buffer containing 1× tricaine. Live imaging, photo-conversion, and opto-activation were performed using a Leica SP5 confocal microscope with a ×40 water immersion objective. Photo-conversions were applied to manually selected regions of interest (ROIs) with a 405 nm laser for 10–20 seconds to induce conversion, monitored by simultaneous imaging at 488 nm to ensure complete conversion. For optogenetic activation, the same ROI was continuously exposed to a 450 nm laser during live imaging to stimulate targeted cellular responses. Sequential imaging of Venus and GFP is described in Figure S11.

## Image analysis and Data analysis

Images were analyzed using ImageJ 2.9.0 and MATLAB 2023b with homemade code. Z-stacks were flattened by maximum or sum slice intensity projections in the ImageJ software. Particle Image Velocimetry (PIV) was employed to measure velocity fields in fluid flow, utilizing a custom MATLAB script based on auto-correlation. PIV captures consecutive images of Cdh5 clusters, and particle displacements between frames are calculated to determine velocity vectors. The homemade MATLAB script processes these images by performing auto-correlation within interrogation windows, extracting particle displacements, and generating velocity field data. The accuracy of the script was validated against known Cdh5 detachment patterns to ensure reliable measurements. Relative signal levels of Cdh5, Myl9a, and Rasip1 at the boundary and apical regions were quantified using MATLAB and custom scripts, employing the freehand selection tool. The boundary selection process involves a hybrid approach that combines automated analysis with manual input from the user. Users interact with the script through a graphical interface, where they can manually select and adjust boundaries using drawing tools. The boundary region is defined as a fixed length of 0.5 μm from the periphery selected by users. Image thresholding was performed using an automatic thresholding method based on Otsu's algorithm, implemented through the graythresh function in MATLAB. This approach determines an optimal global threshold by maximizing the inter-class

variance between the foreground and background pixels. The computed threshold value was then applied to binarize the grayscale images, segmenting regions of interest for further analysis. Automatic thresholding was applied to local regions within each channel using MATLAB and custom scripts. The coefficient of variation was calculated in ImageJ using the freehand selection and measurement functions. The Dice-Sørensen coefficient was measured using custom MATLAB scripts, with freehand selection for areas of interest and automatic thresholding. The above custom scripts are available in the Code Ocean repository linked with this submission. Image panels were created using the Open Microscopy Environment (OMERO).

## Statistics and reproducibility

GraphPad Prism 9 and MATLAB 2023b were used to perform statistical analyses. Immunostaining for Rasip1 and Cdh5, Rasip1 and Myl9a-GFP as well as Rasip1 and ZO-1, were successfully replicated at least three times in independent experimental setups. Live imaging experiments involving Rasip1 and Cdh5, Rasip1 and Myl9a, Rasip1 and UCHD, as well as Cdh5 and Myl9a, were successfully replicated a minimum of five times in both wild-type and mutant embryos, where applicable, across independent experimental setups, yielding consistent results. Drug treatment, Opto-RhoA experiments and the expression of N-ROCK1-Scarlet-Podxl1 were successfully replicated at least three times in independent experimental setups. Results were consistent across all replicates, confirming the robustness and reproducibility of the findings. Sample sizes were not pre-determined using statistical methods, and all data meeting the quality control criteria were included in the analyses. Randomization was not applied to the experiments. Unless specified otherwise, default settings were used for all software tools in the analyses. Whenever a p-value is presented in the text or figures, the corresponding statistical test is indicated.

## Immunofluorescence

For immunofluorescence and imaging of zebrafish embryos, dechorionated embryos were fixed in 2% paraformaldehyde and 0.1% Tween 20 in PBS (PBST) overnight at 4 °C unless otherwise stated. For immunofluorescence involving Rasip1, dechorionated embryos were fixed in 2% TCA in PBST for 15 minutes at room temperature. After fixation, embryos were washed four times with PBST and permeabilized with 0.5% Triton X-100 in PBS at room temperature (RT) for 30 minutes. They were then blocked with 2% BSA and 5% goat serum in PBST overnight with continuous shaking. The following antibodies were used: rabbit anti-zf-Podxl (1:200)[10], rabbit anti-zf-Cdh5 (1:200)[9], guinea pig anti-zf-Cdh5 (1:200)[59], rabbit anti-Rasip1 (1:500)[20], mouse anti-human ZO-1 (1:500; Invitrogen, # 33-9100), Alexa 568 goat anti-rabbit immunoglobulin G (IgG) (1:1000; Thermo Fisher Scientific, A11011), Alexa 488 goat anti-mouse IgG (H + L) (1:1000; Thermo Fisher Scientific, A32723), and Alexa 488 goat anti-guinea pig IgG (H + L) (1:1000; Thermo Fisher Scientific, A11073). Embryos were incubated with primary and secondary antibodies at 4 °C overnight with continuous shaking and were washed six times in between. After immunostaining, embryos were mounted in 0.75% low-melting-point agarose, and images were taken with a Leica SP5 using a 40x water immersion objective.

## Chemical treatment

Embryos were incubated in E3 medium containing 75 μM Y-27632 (MCE, HY-10583) or 1% DMSO as controls. The same concentrations of chemicals were applied to the low-melting-point agarose mounting medium and the E3 medium on top of it before imaging.

## Reporting summary

Further information on research design is available in the Nature Portfolio Reporting Summary linked to this article.

## Data availability

All data supporting the findings of this study are available on Switchdrive: https://drive.switch.ch/index.php/s/UpAJZDe0zVwt8oF. The datasets include Z-projected images, processed data, and analysis scripts. Raw imaging files can be obtained from the corresponding author upon reasonable request. Source data are provided with this paper.

## Code availability

The code utilized for data analysis in this study is available on Code Ocean: https://doi.org/10.24433/CO.0511631.v1. This repository includes custom scripts developed for image processing, quantification, and statistical analysis. The code and associated datasets can also be accessed via this link: https://drive.switch.ch/index.php/s/UpAJZDe0zVwt8oF. For further inquiries, please contact the corresponding author, Jianmin Yin, at jianmin.yin@unibas.ch.

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

## Acknowledgements

We thank Kumuthini Kulendra for fish care and the Imaging Core Facility of the Biozentrum (University of Basel) for microscopy support. We thank Hiroyuki Nakajima (NCVC, Osaka, Japan) for providing *Tg(fli1a: GFP-Podxl1)^{ncv530Tg}*. We thank Anne Schmid (Institute Pasteur, Paris, France) for

providing *Tg(kdrl: Myl9a-GFP)*[ip5Tg]. We thank the Michel Bagnat (Duke University, USA) for providing *TgKI(tjp1a-tdTomato)*[pd1224]. We thank Salim Seyfried (Universität Potsdam, Germany) for providing *heg1*[m552] mutants. We thank Bettina Kirchmaier (Goethe-Universität, Frankfurt, Germany) for providing *krit1*[t26458] mutants. This work has been supported by the Kantons Basel-Stadt and Basel-Land and by grants from the Swiss National Science Foundation (310030_200701 and 310030B_176400) to M.A.

## Author contributions

J.Y., H.-G.B. and M.A. designed the experiments. J.Y. performed the experiments and analyzed the data. J.Y. and M.L. analyzed the *rasip1* mutant phenotype. N.S., K.G. and J.Y. generated the *Tg(cdh5: cdh5-mClavGR2)*[ubs58]. J.Y., L.M. and M.P.K generated the *fli1a: RhoA-BcLOV4-mCherry* construct. L.-K.P. generated *Tg(fli:gal4; UAS: Myl9b-mCherry)*[rk32]. J.Y., C.W. and L.M. analyzed Myl9a distribution. J.Y., H.-G.B. and M.A. wrote the manuscript.

## Competing interests

The authors declare no competing interests.
