## [Peer Review file · Nature Communications]

Initiation of Lumen Formation from Junctions via Differential Actomyosin Contractility Regulated by Dynamic Recruitment of Rasip1

Corresponding Author: Dr Jianmin YIN

Version 0:

Reviewer comments:

Reviewer #1

(Remarks to the Author)

The manuscript by Yin and colleagues demonstrates a role for Rasip1 in establishing de novo vascular lumen formation by controlling junction and lumen integrity. This is achieved by controlling cortical contractility through localized suppression of NMII activity. Interestingly, Rasip1 is demonstrated to move between junctions and the apical compartment, effectively suppressing NMII at the apical site, allowing outward cortical flow that removes Cdh5-junctions from the apical compartment.

A previous report has demonstrated that Rasip1 regulates vascular lumen formation through regulating apical junction clearance, mediated by Rasip1 controlling different pools of GTPases and NMII (PMID: 27486147). Much of the current manuscript by Yin et al. confirms this functional role for Rasip1 in controlling different pools of NMII activity. The main novelties of the current manuscript by Yin et al. are demonstrating that Rasip1 is one of the earliest factors for lumen formation recruited to adhesion sites, and that it dynamically shuttles between the junction and apical sites to regulate myosin contractility. The use of live imaging in to decipher the spatial and temporal events is a strength of the manuscript. However, the main limitation is that the authors do not address two critical aspects related to the mechanism by which Rasip1 functions to control different pools of NMII at junctions and the apical site: How Rasip1 is initially localized to the junction site prior to known apical determinants, and how Rasip1 translocates between the junction and apical compartments to sites of high NMII activity. Additional experiments to identify mechanisms that contribute to the Rasip1 dynamic localization should be included to address this.

Minor comments:

In Figure 6F and G figure legend and main text, UCHD-mRuby is not described. What is this marking?

For figure 4D and F, it is not clear how the radial movement was measured or what the scale represents.

(Remarks on code availability)

Custom code in ImageJ and MATLAB were produced, but this code does not appear to be available.

Reviewer #2

(Remarks to the Author)

General Comments

This study presents a detailed investigation into the intracellular signaling events involving Rasip1 during vascular lumen formation, using zebrafish as a model system. Zebrafish is a unique and potentially ideal model to study this cellular process especially now with many of the molecular toolsets the authors present in the study herein. Along these lines, the authors have developed an impressive array of molecular reporters, photo-convertible tools, and genetic analyses to visualize and manipulate the establishment of apical domains from pre-apical junctional complexes. The identification of Rasip1 as a key player in this process, along with its role in regulating cortical contractility and junctional integrity, represents a significant contribution to our understanding of vascular lumen formation.

Positive Aspects

1. **Innovative Approach:** The use of zebrafish combined with advanced molecular reporters and genetic tools provides a powerful and innovative approach to study the dynamics of lumen formation at a high temporal and developmental resolution.
2. **Comprehensive Analysis:** The study offers a comprehensive analysis of Rasip1's role, detailing its recruitment to apical sites, modulation of contractility, and its dynamic movement between junctional and apical compartments.
3. **Significant Findings:** The identification of Rasip1 as an early apical protein and its involvement in the regulation of cortical contractility and junctional integrity is a notable discovery, adding valuable insights into the mechanisms of vascular lumen formation.

Areas for Improvement

1. **Data Interpretation:** Some results are difficult to interpret from the included data sets. While the study provides a descriptive account of Rasip1's functions, additional robust quantitative data and mechanistic insights would strengthen the conclusions. Specific examples:
 - a. One main conclusion is that Rasip1 regulates differential actomyosin contractility. Fig 5 argues that Rasip1 inhibits contractility junctional patches and apical compartments was difficult to interpret along with graphs in H'. The difference in contractility wasn't robust from the studies presented. A'-C' were difficult to interpret.
 - b. No robust quantitative assessment is included in Fig 1, 4, and 6
2. **Descriptive Nature:** Certain sections of the study remain largely descriptive. Incorporating more functional assays or perturbation experiments to validate Rasip1's mechanisms would enhance the robustness of the findings. For example:
 - a. Concluding model in Fig 8B describes a temporal map of junction, myosin, Rasip1, and Podxl1 temporal and spatial assembly during lumen formation that the paper beautifully lays out. Fig 8C demonstrates how this temporal and spatial map is disrupted with mutant rasip1. Can the authors instead create a model that is clearly supported by experimental data of how Rasip1 defines or is positioned within this temporal and spatial map mechanistically. Right now, its positioned in a way where it messes up overall contractility and junctional formation, but in what way does it specifically do this? Can this be addressed through a model that summarizes the experimental work performed or maybe will highlight experiments that might strengthen the model.
3. **Visual Data:** Given the emphasis on direct visualization, clearer and more consistent image presentation would be beneficial. This includes image color selection consistency, potentially higher resolution images (if possible) or more quantitative image analyses to support the qualitative observations presented (mentioned above for Fig 1,4, 6).
 - a. Specific struggles came from interpreting images with green, red, and blue fluorescence. It might be best to show single channels in black and white to demonstrate the colocalization or clustering that is trying to be emphasized and then in combined images use a color channel. Many of the images were very tiny, and this reviewer struggled to see the result that was being highlighted.

In summary, the study offers valuable contributions and presents an innovative approach to studying vascular lumen formation and the role of Rasip1 in the process. Addressing the issues related to data interpretation and the descriptive nature of some results would significantly improve the manuscript.

(Remarks on code availability)

Reviewer #3

(Remarks to the Author)

Using live imaging of fluorescent and photoconvertible reporters in zebrafish combined with mutant analysis, the authors describe a novel mechanism for an early event involved in de novo vascular lumen formation. Specifically, they identify the clearance of junctional proteins from the newly formed apical membrane to a peripheral junctional ring. The authors show that Rasip1, an upstream regulator of Rho-GTPase activity, is one of the earliest proteins recruited to the apical membrane and is responsible for restricting Cdh5 to junctions by inhibiting myosin contractility at the centre of the apical membrane.

The results described in the manuscripts are novel, experiments well described and the authors' reasoning and hypothesis are easy to follow. However, the imaging and the analysis of the imaging data, which form the sole basis for all the conclusions of the manuscript, approach the limits of credibility due to the limited temporal and spatial resolution in many key figures. I acknowledge that this analysis is technically challenging because the apical membrane clearance occurs within a short timeframe and in a small, spatially restricted location. Many of the questions specified below could be resolved with higher resolution live imaging and strengthening of the key data by additional imaging of fixed, antibody-stained tissue where higher resolution can be obtained compared to live imaging of reporters.

Major comments

1. Most conclusions are drawn from qualitative image data, and it is unclear how many observations were made to support each conclusion. I recognize that providing quantitative data for many of the observations is challenging. Yet, some of the

data are not convincing and it is difficult to fully appreciate the stated observations, see the next point.

2. The convincing findings presented in the study are the localization of Rasip1, its requirement for the compartmentalization of Cdh5 and Podxl1, as well as the rasip1 mutant phenotypes and rescue by ROCK inhibition. However, the data claiming to support local ectopic elevation of actomyosin contractility within the apical compartments in rasip1 mutants as the key mechanistic explanation for the observed phenotypes is not convincingly shown. Specifically, the following statements are not sufficiently supported by the data provided:

- a. Statement in lines 230-231 that the Myl9a-gfp signal largely overlaps with junctional patches in Fig. 5A I & II or in Fig S5. Is the automated thresholding diagrams depicted below performed on the entire image depicted above? Similarly, the initial co-localisation of Myl9 with Rasip1 is not at all clear. Can this data be complemented with immunohistochemical analysis?
- b. "Rasip1 was strongly enriched in clusters together with Myl9a hotspots at junctions or within the apical compartments (Fig. 6I(II) and 6J(I and II), Video 18 and 19)". For example, in J (II) most signals do not seem to co-localize.
- c. Depletion of Myl9a at the junctions and apical membranes in wild type fish (Fig. 6I and 6J), and sustained Myl9a clustering at junctions and within the apical compartments in rasip1 mutants (Fig. 6K and 6L) are not sufficiently supported by the data provided. Additional data showing increased contractility in Rasip1 mutant UCHD-mRuby would strengthen the arguments in line 288-290.
- d. "After continuous 450nm laser activation within the apical compartments in wild-type embryos, Cdh5 clusters emerged at the apical compartment together with the RhoA-BcLOV4-mCherry clusters, while the boundary Cdh5 became fractured and fuzzy (Fig 7H and 7I; Video 20)." The effect seems very limited. Increase in contractility should be shown using the Myl9-reporter.

3. Many of the supplementary figures utilizing antibody staining have a greater resolution and more clearly illustrate the observations than the main figures. The authors should consider swapping certain figures. For example, while the junctional ring formation in Fig. 1B is clearly illustrated both by Cdh5 and ZO1 localization, it is much less clear in Fig. 1C. Supp Fig. 1B, C are much clearer in this regard.

4. Clear labelling of the apical membrane/domains would be helpful for the visualization and localization of apical proteins, especially in figures where quantification is performed within the apical membrane (Fig. 2). The labelling of the apical membrane could be achieved by mosaic expression of a different fluorophore in a subset of endothelial cells (either via plasmid injection or cell transplantation) in an EC reporter background. Neighbouring cells expressing different fluorophores would have an overlapping signal at their apical membrane allowing visualisation of apical membrane formation and expansion. While live imaging of apical membrane of apical proteins in such mosaics might be challenging due to overlapping wavelengths, antibody staining of the various proteins (eg. Rasip1, Podxl, Cdh5, Zo1) should be easily achieved.

5. In Fig. 4A the annotated VE-cadherin particle in wild type fish appears as if very stable during the time period presented in the still images. However, in the video which covers a much larger time frame it is highly mobile. Using the same timepoints in all still images would allow for easier comparison.

6. Myl9a and Cdh5 clustering/co-localisation in 5H' is quantified in a very small ROI and, as with most data, it is unclear on how many samples such analysis was performed. Quantification across the entire apical surface on multiple samples should be performed to justify the claim of co-localization.

7. The statement that ROCK inhibition in wild-type embryos did not alter junctional morphology is not in line with the images presented in Fig. S7 where the treated embryos have much less clearly defined junctional ring and fuzzy Cdh5-positive apical membranes. This is not surprising since ROCK inhibition would not only inhibit actomyosin contractility at the apical surface but also at the junctions where it is critical to recruit Rasip1 (as stated in lines 284-285). Rasip1 reporter fish treated with Y-27632 would show if there is a potential mislocalization under these conditions.

8. Data in Fig. 7E is elegant and convincing but would be even stronger if the authors could show high contractility (Myl9a-GFP) at the apical membrane.

Minor comments:

1. For a broader readership outside the zebrafish field, a larger overview image/video highlighting the area under investigation (DLAV anastomosis and lumen formation) would aid the reader in understanding the exact process the manuscript describes. Ideally in the context of dye injection or erythrocyte labelling to show when lumen formation is initiated and flow is established in these vessels.

2. Many images are depicted in a red-green colour scheme (Fig.1 B,C, Fig.3, Fig.4 A,B,C,F, Fig. 5, Fig. 6, Fig7 C, H, Fig. 8A). Please change to a colour-blind safe LUT such as magenta/green, Cyan/Red or other combinations to ensure the data is fully accessible to all readers.

3. The authors interchangeably perform their analysis in the true DLAV sprouts and in apical membranes localised between DLAV and ISV vessels. The authors are asked to justify this choice or to restrict their analysis to one location of lumen formation.

4. Please adjust Fig2 H, I to show the corresponding images at the same magnification.

5. Line 172: Change the term 'complete conversion' to partial as is confusing with the previous paragraph, where entire

junctional patches were converted.

6. Rephrase line 267-268 to clarify that Rasip1 is within the apical surface in order to not imply that it localises at the junctional rings.

7. Line 273: For broader readership, provide an explanation for what UCHD-mRuby labels since this line is not introduced earlier.

(Remarks on code availability)

Reviewer #4

(Remarks to the Author)

(Remarks on code availability)

Version 1:

Reviewer comments:

Reviewer #1

(Remarks to the Author)

The revisions have addressed my previous concerns. I have no further concerns. This is an excellent paper!

(Remarks on code availability)

Reviewer #2

(Remarks to the Author)

I enjoyed this manuscript and the careful revisions by the authors. The authors addressed all my comments from their first submission and I think this paper should be accepted for publication.

(Remarks on code availability)

Reviewer #3

(Remarks to the Author)

The authors have significantly improved the manuscript by adding high-resolution images of immunofluorescent stainings, which now convincingly support the live imaging data. Additionally, they have made an effort to provide more rigorous and transparent quantitative data. I have only a few remaining minor comments:

1. The authors should avoid using subjective language that overstates their findings. Examples include:
 - a. line 21, 77: 'a wealth of molecular reporters', simply stating novel reporters or a variety of reporters would be sufficient
 - b. line 80: '... advanced live imaging', what is defined as advanced?
 - c. Line 281: '... the massive detachment of cdh5...'
2. Line 135: there are no blue arrows in Fig 2A II
3. Line 136-138: The statement on newly forming and existing apical membranes between tip cells and stalk cells adds essential information, but it would be helpful to also annotate these in Fig. 2B
4. Line 139: It remains unclear what threshold/cut off was used to define boundary area vs apical area in the quantifications of Fig 2B and C. Is this a fixed length from the periphery or based on an intensity threshold?
5. Line 366-371: A small schematic depicting the pathway would help in understanding the paragraph.

(Remarks on code availability)

Reviewer #4

(Remarks to the Author)

(Remarks on code availability)

Reviewer #1 (Remarks to the Author):

The manuscript by Yin and colleagues demonstrates a role for Rasip1 in establishing *de novo*
vascular lumen formation by controlling junction and lumen integrity. This is achieved by controlling
cortical contractility through localized suppression of NMII activity. Interestingly, Rasip1 is
demonstrated to move between junctions and the apical compartment, effectively suppressing NMII
at the apical site, allowing outward cortical flow that removes Cdh5-junctions from the apical
compartment.

We thank the reviewer for the generally positive take on our manuscript.

• A previous report (Barry et al., 2016) has demonstrated that Rasip1 regulates vascular lumen
formation through regulating apical junction clearance, mediated by Rasip1 controlling different pools
of GTPases and NMII (PMID: 27486147). Much of the current manuscript by Yin et al. confirms this
functional role for Rasip1 in controlling different pools of NMII activity. The main novelties of the
current manuscript by Yin et al. are demonstrating that Rasip1 is one of the earliest factors for lumen
formation recruited to adhesion sites, and that it dynamically shuttles between the junction and
apical sites to regulate myosin contractility. The use of live imaging in to decipher the spatial and
temporal events is a strength of the manuscript.

We appreciate the reviewer highlighting the previous work by Barry et al. (2016), which established
the role of Rasip1 in regulating vascular lumen formation through the control of apical junction
clearance, mediated by different pools of GTPases and NMII. Our current manuscript builds upon
this foundational understanding and provides significant advancements in the field of *de novo* lumen
formation. By employing a comprehensive array of molecular reporters, photo-convertible and
optogenetic tools, and genetic analyses, our study offers a comprehensive understanding on *de*
*novo* lumen formation under differential actomyosin contractility controlled by dynamic recruitment
Rasip1. Below are the key contributions of our study:

1. We uncovered the role of differential contractility in *de novo* lumen formation by developing and
utilizing a series of molecular reporters, photoconvertible, and optogenetic tools. These tools allowed
33 us to precisely analyze the dynamics of Cdh5, Myosin, and Rasip1 across various cellular
compartments and to manipulate contractility in specific regions.

2. Our study identified Rasip1 as one of the earliest apical proteins recruited during lumen formation,
dynamically shuttling between junctional and apical sites to regulate myosin contractility, as noted by
the reviewer.

3. We discovered that Rasip1 acts as an inhibitor, rather than an activator, of NMII at the pre-apical
patches. This inhibition creates differential cortical contractility by reducing NMII activity at the
center, which initiates the outward movement of junctional complexes toward the periphery.

4. We revealed a novel mechanism in which Rasip1 contributes to apical clearance by restricting
Cdh5 to the junctions, preventing its detachment by suppressing apical contractility.

These findings offer a deeper and more nuanced understanding of Rasip1's role in *de novo* lumen
formation and junction stabilization, significantly advancing the current knowledge in the field.

• However, the main limitation is that the authors do not address two critical aspects related to the
mechanism by which Rasip1 functions to control different pools of NMII at junctions and the apical
site: How Rasip1 is initially localized to the junction site prior to known apical determinants, and how

Rasip1 translocates between the junction and apical compartments to sites of high NMII activity.
Additional experiments to identify mechanisms that contribute to the Rasip1 dynamic localization
should be included to address this.

We appreciate the reviewer's insightful comments highlighting the critical aspects the dynamic
localization of Rasip1, specifically its recruitment to the initial cell-cell adhesion site and its dynamic
translocation between the junctional and apical compartments. We acknowledge that these are
important questions and agree that understanding the mechanisms behind the dynamic localization
of Rasip1 is crucial for a more comprehensive understanding of its role in vascular development.

To address these points, we propose to include additional experiments aimed at identifying the
factors that contribute to the recruitment of Rasip1 to the initial cell-cell adhesion site and its dynamic
translocation between the junctional and apical compartments. These experiments will involve
Rasip1 immunostaining and live imaging in *heg1*, *krit1(ccm1)* and *cdh5* mutants and live imaging of
Rasip1-Scarlet-I upon opto-RhoA activation (lines 361-395).

One key unanswered question is how Rasip1 achieves such specific and dynamic localizations.
Rasip1 and Krit1 (CCM1), another key regulator of Rho GTPases, bind to Heg1, a transmembrane
protein, at different binding sites of the cytoplasmic domain of Heg1 (de Kreuk et al., 2016; Gingras et
al., 2012). The interactions between Heg1 and Rasip1 had been proposed to facilitate recruitment of
Rasip1, leading to the stabilization of endothelial cell-cell junctions (de Kreuk et al., 2016). It has also
been reported that Krit1 is recruited to cell-cell junctions and forms complexes with Cdh5 (Béraud-
Dufour et al., 2007; Glading et al., 2007; Lampugnani et al., 2010; Liu et al., 2011).

Our previous study demonstrated that *heg1* and *krit1* mutants exhibited distinct apical domains
surrounded by zig-zag or contorted cell-cell junctions, in sharp contrast to the *rasip1* mutants (Yin et
al., 2024). In contrast, junctional rings appeared more circular in *cdh5* mutants (Sauteur et al., 2014).
Despite these distinct shapes of cell-cell junctions, the segregation of apical compartments from
junctions remained largely unaffected in *heg1*, *krit1*, and *cdh5* mutants (Sauteur et al., 2017; Yin et al.,
2024). This allowed us to explore Rasip1's dynamic recruitment in these mutants through Rasip1
immunostaining and live imaging (Fig. 8A-D). Interestingly, the recruitment of Rasip1 to the initial cell-
cell contact sites and apical compartments appeared largely independent of Heg1, Krit1, and Cdh5
(Fig. 8A). However, Rasip1 was poorly localized at junctions in *heg1* and *krit1* mutants, as shown by
antibody staining and live imaging, suggesting that Heg1 and Krit1 are involved in Rasip1's dynamic
translocation between the junctional and apical compartments (Fig. 8B-D; Video 21). It is interesting
to note that Rasip1 was strongly enriched at junctions in *cdh5* mutants, suggesting the involvement of
other junctional proteins in positioning Rasip1 to junctions (Fig. 8B and 8C).

Rasip1 is first localized at the initial cell-cell adhesion sites together with Myl9a (Fig. 5A-C).
Following the formation of stable junctional rings, Rasip1 dynamically translocates to newly formed
Myl9a clusters and constricting junctions (Fig. 6F and 6I(II and III)). These observations suggest that
Rasip1 is dynamically recruited, potentially driven by increased local contractility. To further explore
this hypothesis, we used opto-RhoA-GFP to selectively activate RhoA in the apical compartments or
junctions (Fig. 8E and 8F). We observed dynamic recruitment of Rasip1-Scarlet-I, which colocalized
with RhoA-BcLOV4-GFP clusters within the apical compartments and junctions after activation (n=4)
(Fig. 8E, 8F and S9). These findings suggest that Rasip1 is dynamically recruited in respond to
heightened contractility.

**Minor comments:**

In Figure 6F and G figure legend and main text, UCHD-mRuby is not described. What is this
marking?

We apologize for the oversight in not describing mRuby2-UCHD. The junctional complexes are
associated with the actin cytoskeleton and are therefore visualized using the F-actin binding domain
of utrophin fused to mRuby2 (mRuby2-UCHD) (lines 305-306).

For figure 4D and F, it is not clear how the radial movement was measured or what the scale
represents.

The movement of junctional clusters was analyzed using Particle Image Velocimetry (PIV), an
optical measurement technique used to visualize and analyze fluid flow or movement in various
systems, including biological tissues, by tracking the motion of small particles within the flow. We
applied this method to high temporospatial resolution movies. The PIV code is available in the Code
Ocean repository linked with our submission. Specifically, we calculated the movement of junctional
clusters along radial directions relative to the center of the junctional rings to compare wild-type
embryos with *rasip1* mutants. The heat map colors represent the distance (in μm) of movement
along the radial direction over a 6-minute interval, as indicated by the scale bars. Red indicates
outward flow away from the center, while blue indicates inward flow toward the center (lines 214-
217, 613-621).

Reviewer #1 (Remarks on code availability):
Custom code in ImageJ and MATLAB were produced, but this code does not appear to be available.
The custom code is stored in the Code Ocean repository as cloud-based executable capsule
available to reviewers. Upon acceptance of a manuscript, the final version of the capsule will be
published with a permanent DOI (lines 911-915).

Reviewer #2 (Remarks to the Author):

General Comments

This study presents a detailed investigation into the intracellular signaling events involving Rasip1
during vascular lumen formation, using zebrafish as a model system. Zebrafish is a unique and
potentially ideal model to study this cellular process especially now with many of the molecular
toolsets the authors present in the study herein. Along these lines, the authors have developed an
impressive array of molecular reporters, photo-convertible tools, and genetic analyses to visualize
and manipulate the establishment of apical domains from pre-apical junctional complexes. The
identification of Rasip1 as a key player in this process, along with its role in regulating cortical
contractility and junctional integrity, represents a significant contribution to our understanding of
vascular lumen formation.

Positive Aspects

1. Innovative Approach: The use of zebrafish combined with advanced molecular reporters and
genetic tools provides a powerful and innovative approach to study the dynamics of lumen formation
at a high temporal and developmental resolution.
2. Comprehensive Analysis: The study offers a comprehensive analysis of Rasip1's role, detailing its
recruitment to apical sites, modulation of contractility, and its dynamic movement between junctional
and apical compartments.
3. Significant Findings: The identification of Rasip1 as an early apical protein and its involvement in
the regulation of cortical contractility and junctional integrity is a notable discovery, adding valuable
insights into the mechanisms of vascular lumen formation.

We thank the reviewer for the generally positive take on our manuscript.

Areas for Improvement

1. **Data Interpretation:** Some results are difficult to interpret from the included data sets. While the study provides a descriptive account of Rasip1's functions, additional robust quantitative data and mechanistic insights would strengthen the conclusions. Specific examples:

We appreciate the reviewer's insightful comments. While the manuscript primarily outlines the recruitment and segregation of proteins during apical polarization, along with functional studies on Rasip1 using genetic and optogenetic approaches, we acknowledge the need for additional quantitative analyses and mechanistic insights to further substantiate our conclusions.

To address this, we have taken the following experiments and analyses:

1. To complement our live imaging approaches, we conducted additional immunostainings, imaging and co-localization analyses at various stages of *de novo* lumen formation (Current Fig. 1 C, 1D, 5A and 5B). These efforts enhance the robustness of our quantitative analyses by using images from multiple embryos with improved signal-to-noise ratios. These new data serve as a valuable complement to our live imaging approach (lines 101-108 and 244-247).
2. To gain a deeper mechanistic insight on the dynamic recruitment of Rasip1, specifically its initial localization to the junction site and its dynamic translocation between the junctional and apical compartments, we included additional experiments aimed at identifying the factors that contribute to the initial cell-cell adhesion site and its dynamic translocation between the junctional and apical compartments. These experiments will involve Rasip1 immunostaining and live imaging in *heg1*, *krit1(ccm1)* and *cdh5* mutants and live imaging of Rasip1 upon opto-RhoA activation (Current Fig.8A-F) (lines 361-395).

a. One main conclusion is that Rasip1 regulates differential actomyosin contractility. Fig 5 argues that Rasip1 inhibits contractility junctional patches and apical compartments was difficult to interpret along with graphs in H'. The difference in contractility wasn't robust from the studies presented. A'-C' were difficult to interpret.

We appreciate the reviewer's detailed feedback regarding the interpretation of Fig. 5 and the associated data on Rasip1's regulation of differential actomyosin contractility.

To address the concerns:

1. We performed high-resolution immunostainings of Myl9a-GFP and Rasip1 at various stages, from initial cell-cell contacts to the formation of stable junctional rings (current Fig. 5A). Using the Dice coefficient, we quantified the level of colocalization between the two signals. Our analysis reveals a shift from strong colocalization at initial contacts to segregation in stable junctional rings (current Fig. 5B).
2. We have now placed time-lapse images of Cdh5, Rasip1, and Myl9a in wild-type conditions as current Fig. 5C and D after the new immunostaining data. These live imaging data complement the immunostainings by clearly illustrating the sequence of rearrangements.
3. We have revised current Fig. 5C and 5E with high-resolution samples and an extended time window to improve clarity and better convey the differences in junction morphology and actomyosin contractility between wild-type embryos and *rasip1* mutants.
4. The different actomyosin contractility between wild-type embryos and *rasip1* mutants was presented and quantified in current Fig. 5F-H.
5. We have added a new Figure 5K. Along with the current Figures 5I and 5J, we demonstrate the dynamics of Myosin and junctional remodeling in both wild-type embryos and *rasip1*

mutants, accompanied by new diagrams in **Figures 5I'-K'**, and number of observations in the
main text.

**b. No robust quantitative assessment is included in Fig 1, 4, and 6**

We appreciate the reviewer's feedback. In this study, we primarily used live imaging approaches to
resolve the sequential events during *de novo* lumen formation. Quantifying live imaging data is
challenging and can sometimes oversimplify valuable dynamic processes. In response, we have
now included robust quantitative assessments on fixed samples with a large sample size.

1. **Current Fig 1C and 1D:** we performed immunostainings and co-localization analyses on
Cdh5 and Rasip1 at various stages of *de novo* lumen formation.
- 2. **Current Fig 5A and 5B:** we performed immunostainings and co-localization analyses on
Myl9a and Rasip1 at various stages of *de novo* lumen formation.
- 3. **Current Fig 4F:** we quantified the average radial velocity of Cdh5 clusters in both wild-type
embryos and *rasip1* mutants, revealing the difference in the radial velocity between the two
groups, as well as the pulsatile dynamics observed in *rasip1* mutants.
- 4. **Current Fig 8C:** we quantified the relative boundary-to-apical ratio of Rasip1 in fixed wild-type
embryos and various mutants, serving as a complementary quantitative assessment for the
dynamics of Rasip1 in **Figure 6**.

**2. Descriptive Nature: Certain sections of the study remain largely descriptive. Incorporating more**
**functional assays or perturbation experiments to validate Rasip1's mechanisms would enhance the**
**robustness of the findings. For example:**

**a. Concluding model in Fig 8B describes a temporal map of junction, myosin, Rasip1, and Podx1**
**temporal and spatial assembly during lumen formation that the paper beautifully lays out. Fig 8C**
**demonstrates how this temporal and spatial map is disrupted with mutant rasip1. Can the authors**
**instead create a model that is clearly supported by experimental data of how Rasip1 defines or is**
**positioned within this temporal and spatial map mechanistically. Right now, its positioned in a way**
**where it messes up overall contractility and junctional formation, but in what way does it specifically**
**do this? Can this be addressed through a model that summarizes the experimental work performed**
**or maybe will highlight experiments that might strengthen the model.**

Thank you for your insightful comments and feedback. To gain a deeper mechanistic insight on the
dynamic recruitment of Rasip1, specifically its initial localization to the initial cell-cell adhesion site
and its dynamic translocation between the junctional and apical compartments, we propose to
include additional experiments aimed at identifying the factors that contribute to the dynamic
recruitment of Rasip1. These experiments will involve Rasip1 immunostaining and live imaging in
*heg1*, *krit1(ccm1)* and *cdh5* mutants and live imaging of Rasip1-Scarlet-I upon opto-RhoA activation
(**Current Fig. 8A-F**) (lines 361-395).

One key unanswered question is how Rasip1 achieves such specific and dynamic localizations.
Rasip1 and Krit1 (CCM1), another key regulator of Rho GTPases, bind to Heg1, a transmembrane
protein, at different binding sites of the cytoplasmic domain of Heg1 (de Kreuk et al., 2016; Gingras et
al., 2012). The interactions between Heg1 and Rasip1 had been proposed to facilitate recruitment of
Rasip1, leading to the stabilization of endothelial cell-cell junctions (de Kreuk et al., 2016). It has also
been reported that Krit1 is recruited to cell-cell junctions and forms complexes with Cdh5 (Béraud-
Dufour et al., 2007; Glading et al., 2007; Lampugnani et al., 2010; Liu et al., 2011).

Our previous study demonstrated that *heg1* and *krit1* mutants exhibited distinct apical domains
surrounded by zig-zag or contorted cell-cell junctions, in sharp contrast to the *rasip1* mutants (Yin et
al., 2024). In contrast, junctional rings appeared more circular in *cdh5* mutants (Sauter et al., 2014).

Despite these distinct shapes of cell-cell junctions, the segregation of apical compartments from
junctions remained largely unaffected in *heg1*, *krit1*, and *cdh5* mutants (Sauteur et al., 2017; Yin et al.,
2024). This allowed us to explore Rasip1's dynamic recruitment in these mutants through Rasip1
immunostaining and live imaging (Fig. 8A-D). Interestingly, the recruitment of Rasip1 to the initial cell-
cell contact sites and apical compartments appeared largely independent of Heg1, Krit1, and Cdh5
(Fig. 8A). However, Rasip1 was poorly localized at junctions in *heg1* and *krit1* mutants, as shown by
antibody staining and live imaging, suggesting that Heg1 and Krit1 are involved in Rasip1's dynamic
translocation between the junctional and apical compartments (Fig. 8B-D; Video 21). It is interesting
to note that Rasip1 was strongly enriched at junctions in *cdh5* mutants, suggesting the involvement of
other junctional proteins in positioning Rasip1 to junctions (Fig. 8B and 8C).

Rasip1 is first localized at the initial cell-cell adhesion sites together with Myl9a (Fig. 5A-C).
Following the formation of stable junctional rings, Rasip1 dynamically translocates to newly formed
Myl9a clusters and constricting junctions (Fig. 6F and 6I(II and III)). These observations suggest that
Rasip1 is dynamically recruited, potentially driven by increased local contractility. To further explore
this hypothesis, we used opto-RhoA-GFP to selectively activate RhoA in the apical compartments or
junctions (Fig. 8E and 8F). We observed dynamic recruitment of Rasip1-Scarlet-I, which colocalized
with RhoA-BcLOV4-GFP clusters within the apical compartments and junctions after activation (n=4)
(Fig. 8E, 8F and S9). These findings suggest that Rasip1 is dynamically recruited in respond to
heightened contractility.

Based on the new mechanistic insights into the dynamic recruitment of Rasip1, we have updated the
working model in the current Figure 9. We have incorporated the interactions between the key
players involved in *de novo* lumen formation by showing that Rasip1 is recruited in response to high
contractility and act to suppress it. Specifically, our findings indicate that the dynamic shuttling of
Rasip1 between junctions and the apical compartment is dependent on Heg1 and Krit1. Additionally,
we have highlighted the gap in understanding regarding how Rasip1 is localized to the cell-cell
contact sites and the apical domains (lines 452-462, 482-491).

3. Visual Data: Given the emphasis on direct visualization, clearer and more consistent image
presentation would be beneficial. This includes image color selection consistency, potentially higher
resolution images (if possible) or more quantitative image analyses to support the qualitative
observations presented (mentioned above for Fig 1,4, 6).

We have improved of image color consistency and color-blind safe LUT in the revised manuscript.
The updated color scheme is as follows: Cdh5 and ZO-1 are mostly represented in green, Rasip1
and Podxl primarily in magenta, and Myl9a in red.

We appreciate your feedback regarding the signal-to-noise ratio and spatial resolution in our live
imaging results. Several factors contributed to these limitations. First, to minimize potential over-
expression artifacts, we specifically selected embryos with lower levels of fluorescent protein
expression. Additionally, the imaging system's speed and the necessity to capture over 100 frames
for dynamic processes constrained the temporal and spatial resolutions. Other contributing factors
include phototoxicity and photobleaching, which can impact image quality and signal stability. We
are continuously working to optimize our imaging conditions and appreciate your understanding of
these challenges.

To address the concerns on the resolution and size of samples quantified, we performed
immunostainings and high-resolution imaging on fixed samples with a large sample size, as we
mentioned before.

a. Specific struggles came from interpreting images with green, red, and blue fluorescence. It might

be best to show single channels in black and white to demonstrate the colocalization or clustering
that is trying to be emphasized and then in combined images use a color channel. Many of the
images were very tiny, and this reviewer struggled to see the result that was being highlighted.

Thank you for your valuable feedback regarding the interpretation of the fluorescence images and
we appreciate your suggestions for improving clarity. To address these concerns, we will revise the
figures as follows:

- 1. We have improved color consistency and color-blind safe LUT in the revised manuscript. The
updated color scheme is as follows: Cdh5 and ZO-1 are mostly represented in green, Rasip1
and Podxl primarily in magenta, and Myl9a in red. The use of blue has been avoided in the
current manuscript.
- 2. Single-Channel Images: We now provide black-and-white versions of the single fluorescence
channels to clearly demonstrate colocalization and clustering in **current Fig 6D-G**.
- 3. We performed automatic thresholding on different channels of fluorescence images in
**current Fig. 2E, 5C-E, S4A, S10**
- 4. We measured and plotted intensities of fluorescent signals along dashed lines to
demonstrate the differences in **current Fig. 1B, 1F, 2D-F, 3E-G, 5F, 5G, 7F, 7H**

In summary, the study offers valuable contributions and presents an innovative approach to studying
vascular lumen formation and the role of Rasip1 in the process. Addressing the issues related to
data interpretation and the descriptive nature of some results would significantly improve the
manuscript.

Thank you for your positive feedback and for recognizing the contributions of our study in advancing
the understanding of vascular lumen formation and the role of Rasip1. We believe these revisions
will enhance the manuscript and provide a more robust understanding of the results. Thank you
again for your valuable suggestions.

**Reviewer #3 (Remarks to the Author):**

Using live imaging of fluorescent and photoconvertible reporters in zebrafish combined with mutant
analysis, the authors describe a novel mechanism for an early event involved in de novo vascular
lumen formation. Specifically, they identify the clearance of junctional proteins from the newly formed
apical membrane to a peripheral junctional ring. The authors show that Rasip1, an upstream
regulator of Rho-GTPase activity, is one of the earliest proteins recruited to the apical membrane
and is responsible for restricting Cdh5 to junctions by inhibiting myosin contractility at the centre of
the apical membrane.

The results described in the manuscripts are novel, experiments well described and the authors'
reasoning and hypothesis are easy to follow. However, the imaging and the analysis of the imaging
data, which form the sole basis for all the conclusions of the manuscript, approach the limits of
credibility due to the limited temporal and spatial resolution in many key figures. I acknowledge that
this analysis is technically challenging because the apical membrane clearance occurs within a short
timeframe and in a small, spatially restricted location. Many of the questions specified below could
be resolved with higher resolution live imaging and strengthening of the key data by additional
imaging of fixed, antibody-stained tissue where higher resolution can be obtained compared to live
imaging of reporters.

We thank the reviewer for the generally positive take on our manuscript. We appreciate your
feedback regarding the signal-to-noise ratio and spatial resolution in our live imaging results. Several
factors contributed to these limitations. First, to minimize potential over-expression artifacts, we
specifically selected embryos with lower levels of fluorescent protein expression. Additionally, the
imaging system's speed and the necessity to capture over 100 frames for dynamic processes
constrained the temporal and spatial resolutions. Other contributing factors include phototoxicity and
photobleaching, which can impact image quality and signal stability. We are continuously working to
optimize our imaging conditions and appreciate your understanding of these challenges.

To address the concerns on the resolution and size of samples quantified, we performed antibody-
based immunostainings and high-resolution imaging on fixed samples with a large sample size
**Current Fig 1C, 1D, 5A, 5B, 8A and 8B**. By combining live imaging with immunostaining, we hope to
obtain a richer, more nuanced view of cellular dynamics by integrating real-time observations with
detailed, high-resolution images with enough sample size accounting for variability between
embryos.

Major comments

1. Most conclusions are drawn from qualitative image data, and it is unclear how many observations
were made to support each conclusion. I recognize that providing quantitative data for many of the
observations is challenging. Yet, some of the data are not convincing and it is difficult to fully
appreciate the stated observations, see the next point.

We appreciate the reviewer's feedback. In this study, we primarily used live imaging approaches to
resolve the sequential events during *de novo* lumen formation. Quantifying live imaging data is
challenging and can sometimes oversimplify valuable dynamic processes. In response, we have
now included robust quantitative assessments on fixed samples with a large sample size.

- 1. **Current Fig 1C and 1D**: we performed immunostainings and co-localization analyses on
Cdh5 and Rasip1 at various stages of *de novo* lumen formation (n=42).
- 2. **Current Fig 5A and 5B**: we performed immunostainings and co-localization analyses on
Myl9a and Rasip1 at various stages of *de novo* lumen formation (n=27).
- 3. **Current Fig 8C**: we performed immunostainings and measured the boundary-to-apical ratio
of Rasip1 in wild-type embryos and various mutants (WT embryos (n=49), *heg1^{m552}* (n=33),
*krit1^{t26458}* (n=57) and *cdh5^{ubs8}* mutants (n=47))
- 4. **Current Fig 5J, 5K, 6I, 6J, 8E, 8F**: we provided the number of experiments or observations.

2. The convincing findings presented in the study are the localization of Rasip1, its requirement for
the compartmentalization of Cdh5 and Podxl1, as well as the rasip1 mutant phenotypes and rescue
by ROCK inhibition. However, the data claiming to support local ectopic elevation of actomyosin
contractility within the apical compartments in rasip1 mutants as the key mechanistic explanation for
the observed phenotypes is not convincingly shown. Specifically, the following statements are not
sufficiently supported by the data provided:

a. Statement in lines 230-231 that the Myl9a-gfp signal largely overlaps with junctional patches in
Fig. 5A I & II or in Fig S5. Is the automated thresholding diagrams depicted below performed on the
entire image depicted above? Similarly, the initial co-localisation of Myl9 with Rasip1 is not at all
clear. Can this data be complemented with immunohistochemical analysis?

The automated thresholding was performed on the entire image depicted in **old Fig. 5A**. Some of the
Myl9a clusters away from the junctional patches were removed for clarity after automated
thresholding.

We appreciate the reviewer's detailed feedback regarding the interpretation of Fig. 5 and the associated data on Rasip1's regulation of differential actomyosin contractility.

To address the concerns:

1. We performed immunostainings on Myl9a-GFP and Rasip1 at various stages, from initial cell-cell contacts to the formation of stable junctional rings (current Fig. 5A). Using the Dice coefficient, we quantified the level of colocalization between the two signals. Our analysis reveals a shift from strong colocalization at initial contacts to segregation in stable junctional rings (current Fig. 5B).
2. We have now placed time-lapse images of Cdh5, Rasip1, and Myl9a in wild-type conditions as current Fig. 5C and D after the new immunostaining data. These live imaging data complement the immunostainings by clearly illustrating the sequence of rearrangements.
3. We have revised current Fig. 5C and 5E with high-resolution samples and an extended time window to improve clarity and better convey the differences in junction morphology and actomyosin contractility between wild-type embryos and *rasip1* mutants.

b. "Rasip1 was strongly enriched in clusters together with Myl9a hotspots at junctions or within the apical compartments (Fig. 6I(II) and 6J(I and II), Video 18 and 19)". For example, in J (II) most signals do not seem to co-localize.

we acknowledge not all Rasip1 and Myl9a clusters co-localized in Figure 6I and 6J. There are several reasons account for that:

1. The dynamic nature of the molecular clusters. The localizations of Rasip1 and Myl9a are very dynamic with new clusters form while old clusters disappear rapidly. The newly formed Myl9a cluster, as shown in current Fig. 6I(II), was not colocalized with Rasip1 at 05:23 (arrowhead). However, Rasip1 was recruited to this newly formed cluster at 07:56 (arrow in current Fig. 6I(III)).
2. While Rasip1 is largely restricted within the apical compartments and junctions, Myl9a localizes throughout the whole cell. Thus, Myl9a outside of the apical compartments and junctions will not co-localize with Rasip1.

To address the concerns:

1. We have extended the time window in Fig. 6I and 6J to better illustrate the dynamic behavior of the molecular clusters, and we have clarified these dynamics in the main text (lines 315-318).
2. We have applied masks to the image panels to clearly label the apical compartments, enhancing the clarity of the localization data.

c. Depletion of Myl9a at the junctions and apical membranes in wild type fish (Fig. 6I and 6J), and sustained Myl9a clustering at junctions and within the apical compartments in *rasip1* mutants (Fig. 6K and 6L) are not sufficiently supported by the data provided. Additional data showing increased contractility in Rasip1 mutant UCHD-mRuby would strengthen the arguments in line 288-290.

We agree with the reviewer that additional data using mRuby2-UCHD would strengthen our claims. To address this, we have conducted further experiments to assess actin localization with mRuby2-UCHD in both wild-type embryos and *rasip1* mutants. The new data reveal ectopic localization of mRuby2-UCHD in the apical compartments of *rasip1* mutants, further supporting our conclusions.

These results are now included in Supplementary Fig. S6A-D. We have relocated these data to the supplementary figures because Fig. 6 primarily focuses on the dynamic localization of Rasip1 in

wild-type embryos. The data shown in **Supplementary Fig. S6A-D** complement **the current Fig. 5F-K**
for showing increased contractility in *rasip1* mutants.

480 d. “After continuous 450nm laser activation within the apical compartments in wild-type embryos,
Cdh5 clusters emerged at the apical compartment together with the RhoA-BcLOV4-mCherry
clusters, while the boundary Cdh5 became fractured and fuzzy (Fig 7H and 7I; Video 20).” The effect
seems very limited. Increase in contractility should be shown using the Myl9-reporter.

We appreciate the reviewer’s constructive feedback on the effects of optogenetic RhoA activation on
Cdh5 clustering and the suggestion for additional evidence of increased contractility. As noted, the
effect of opto-RhoA on Cdh5 localization is indeed milder compared to N-Rock1-Scarlet-Podxl. We
agree that demonstrating contractility using a Myl9a reporter provides more direct evidence of the
contractile response.

In response to this, we conducted new experiments using Cdh5 or Myl9a reporters to assess
contractility following 450nm laser activation in wild-type embryos expressing opto-RhoA. The opto-
activation induced clear Myl9a clustering alongside clustered opto-RhoA-mCherry, confirming
increased contractility in response to RhoA activation (**Supplementary Fig. S8B**).

3. Many of the supplementary figures utilizing antibody staining have a greater resolution and more
clearly illustrate the observations than the main figures. The authors should consider swapping
certain figures. For example, while the junctional ring formation in Fig. 1B is clearly illustrated both
by Cdh5 and ZO1 localization, it is much less clear in Fig. 1C. Supp Fig. 1B, C are much clearer in
this regard.

We thank the reviewer for their valuable feedback regarding the resolution and clarity of the figures.
We agree that the supplementary figures utilizing antibody staining provide greater resolution and
more clearly illustrate the observations. In response to the reviewer’s suggestion, we have added or
revised the current **Fig. 1C, 2A, 5A, 8A and 8B**.

4. Clear labelling of the apical membrane/domains would be helpful for the visualization and
localization of apical proteins, especially in figures where quantification is performed within the apical
membrane (Fig. 2). The labelling of the apical membrane could be achieved by mosaic expression of
a different fluorophore in a subset of endothelial cells (either via plasmid injection or cell
transplantation) in an EC reporter background. Neighbouring cells expressing different fluorophores
would have an overlapping signal at their apical membrane allowing visualisation of apical
membrane formation and expansion. While live imaging of apical membrane of apical proteins in
such mosaics might be challenging due to overlapping wavelengths, antibody staining of the various
proteins (eg. Rasip1, Podxl, Cdh5, Zo1) should be easily achieved.

We thank the reviewer for the valuable suggestion regarding clearer labeling of the apical membrane
to improve the visualization and localization of apical proteins. As noted, we have tried several
methods to label the apical domains in our study:

1. **Apical Markers:** We utilized apical markers such as Rasip1 and Podxl1 to identify the apical
regions.
2. **Junctional Markers:** Given the stereotypical *de novo* lumen formation in DLAV, where
apical membranes are situated within junctional rings, we employed junctional markers like
ZO-1, Cdh5, and the junction-associated actin marker UCHD.
3. **Antibody Staining:** We used antibodies against Rasip1, ZO1, and Cdh5 to label the apical
domains, which provided detailed visualization.

4. **Mosaic Labeling:** We acknowledge that mosaic labeling could enhance the clarity of
antibody staining, as suggested by the reviewer. However, this technique presents
challenges for our live imaging because it requires multiple imaging channels and the
simultaneous presence of two neighboring cells with differential expression.

5. In Fig. 4A the annotated VE-cadherin particle in wild type fish appears as if very stable during the
time period presented in the still images. However, in the video which covers a much larger time
frame it is highly mobile. Using the same timepoints in all still images would allow for easier
comparison.

We appreciate the reviewer's concern regarding the use of consistent timepoints for the still images.
Due to variations in tissue thickness and the number of samples imaged simultaneously, it is
challenging to ensure the exact same time interval for all still images. However, we have taken care
to present data within similar time ranges to ensure comparability. For Fig. 4A-C, we have
consistently shown data within the 400-600 seconds time window.

Indeed, VE-cadherin particles are more mobile later in the movie compared to the beginning. We
believe this reflects the dynamics of the junctional particles over different time periods, which we
also observed and described in *rasip1* mutants. However, due to the very small number of such
particles existed in wild-type embryos, we cannot make further conclusion upon that.

6. Myl9a and Cdh5 clustering/co-localisation in 5H' is quantified in a very small ROI and, as with
most data, it is unclear on how many samples such analysis was performed. Quantification across
the entire apical surface on multiple samples should be performed to justify the claim of co-
localization.

We thank the reviewer for raising the issue regarding the quantification of Myl9a and Cdh5
clustering/co-localization. However, the images in the original Fig. 5H were intended to capture
dynamic events, specifically the formation of ectopic Cdh5 clusters in conjunction with ectopic Myl9a
activity. Given the highly dynamic nature of these clusters, they may not remain co-localized at later
stages.

Quantifying all clusters across the entire image would not accurately reflect the sequential
recruitment process we are focusing on. Our analysis highlights the early stages of clustering, where
Myl9a and Cdh5 initially co-localize, demonstrating the recruitment sequence. We have now added
the number of observations, showing this recruitment pattern in 7 out of 9 observations (lines 279-
281). Furthermore, we observed the collapse of junctions into high-density patches, with persisted
enrichment of Myl9a along junctions as dense clusters in 3 out of 10 *rasip1* mutants (lines 284-288),
suggesting a disruption in the organization of contractility also at the boundary region. To further
clarify the sequence of rearrangement, we have included diagrams illustrating the clustering
sequence, now presented as current Fig. 5I-K.

7. The statement that ROCK inhibition in wild-type embryos did not alter junctional morphology is not
in line with the images presented in Fig. S7 where the treated embryos have much less clearly
defined junctional ring and fuzzy Cdh5-positive apical membranes. This is not surprising since
ROCK inhibition would not only inhibit actomyosin contractility at the apical surface but also at the
junctions where it is critical to recruit Rasip1 (as stated in lines 284-285). Rasip1 reporter fish treated
with Y-27632 would show if there is a potential mislocalization under these conditions.

We thank the reviewer for the comment. Upon careful examination, we did not observe less clearly
defined junctional rings in the embryos transiently treated with the ROCK inhibitor Y-27632. The

discrepancy in interpretation may be due to natural variation in Cdh5 expression levels between
embryos. We reviewed additional frames from the images in Fig. S7 (Reply to Reviewer Figure 1A
and 1B), as well as other samples (Current Figure 7C, Reply to Reviewer Figure 1C and 1D), and
consistently observed well-defined junctional boundaries.

Reply to Reviewer Figure 1.

Rather than focusing on Y-27632 treatment, which could broadly affect both junctional elongation
and contraction, we opted to study Rasip1 localization in *krit1* and *heg1* mutants. These mutants
specifically lack junctional contractility, allowing us to better understand the localization of Rasip1
under conditions where junctional contractility is disrupted (lines 361-385).

We will remove the claim that ROCK inhibition in wild-type embryos did not alter junctional
morphology, since Y-27632 treatment could broadly affect both junctional elongation and contraction
and it is beyond the scope of this study to test the differential effects of this drug on junction
morphology.

8. Data in Fig. 7E is elegant and convincing but would be even stronger if the authors could show
high contractility (Myl9a-GFP) at the apical membrane.

We appreciate the reviewer's constructive feedback on the effects of N-Rock1-Scarlet-Podxl1. In
response to this, we conducted new experiments using Myl9a reporters to assess contractility
expression of N-Rock1-Scarlet-Podxl1. Myl9a clustering was observed alongside N-Rock1-Scarlet-
Podxl1 at the apical compartments after activation, further confirming increased contractility in
response to ROCK activation (Supplementary Fig. S8A).

Minor comments:

1. For a broader readership outside the zebrafish field, a larger overview image/video highlighting
the area under investigation (DLAV anastomosis and lumen formation) would aid the reader in
understanding the exact process the manuscript describes. Ideally in the context of dye injection or
erythrocyte labelling to show when lumen formation is initiated and flow is established in these
vessels.

We appreciated the reviewer's suggestion on larger overview image/video highlighting the area
under investigation. The **current Video1** demonstrates the establishment of junctions and apical
domains between multiple sprouts of DLAV, the exact process of *de novo* lumen formation this
manuscript described. In contrast, showing blood flow-driven lumen formation at later stage could be
distracting, as our primary focus in this study is on *de novo* lumen formation.

2. Many images are depicted in a red-green colour scheme (Fig.1 B,C, Fig.3, Fig.4 A,B,C,F, Fig. 5,
Fig. 6, Fig7 C, H, Fig. 8A). Please change to a colour-blind safe LUT such as magenta/green,
Cyan/Red or other combinations to ensure the data is fully accessible to all readers.

We sincerely thank the reviewer for their valuable suggestion regarding the use of a color-blind safe
LUT. We recognize the importance of ensuring that our figures are accessible to all readers. In
response, we have revised most of the figures and videos to incorporate a color-blind safe LUT.
However, other factors such as maintaining color consistency, clarity, and the complexity of images
with three or more channels had to be considered. For panels that do not fully meet the color-blind
safe LUT requirements, we have provided detailed labels, outlines, and descriptions to ensure clarity
and understanding for all readers, regardless of color perception.

3. The authors interchangeably perform their analysis in the true DLAV sprouts and in apical
membranes localised between DLAV and ISV vessels. The authors are asked to justify this choice or
to restrict their analysis to one location of lumen formation.

We appreciate the reviewer's comment. In our study, we investigated both the initiation of lumen
formation and the maintenance of established apical domains. Specifically, we examined the apical
membrane between tip cells to study the initiation of lumen formation and junctional remodeling
during this process, as illustrated. In contrast, the apical domains between tip cell and stalk cells
were analyzed to explore junctional and apical maintenance, providing a more comprehensive view
of lumen formation across different regions. Additionally, we observed similar morphology of
junctions, molecular localization, and mutant phenotypes at junctions/apical domains in both regions.
The purpose of each experiment, and the types of apical domains used are now properly described
in the current manuscript (**lines 90, 103, 166, 181-182...**) along with a comparison of the phenotype
and molecular localization at junctional rings in both regions (**lines 136-138, 265-268**).

4. Please adjust Fig2 H, I to show the corresponding images at the same magnification.

The different magnifications were chosen to provide both a broader overview and sufficient detail.
Specifically, we wanted to highlight the ectopic presence of Podxl outside of junctional patches in
*rasip1* mutants (**current Fig. 2I**).

5. Line 172: Change the term 'complete conversion' to partial as is confusing with the previous
paragraph, where entire junctional patches were converted.

Thank you for pointing this out. We agree that the term 'complete conversion' could be confusing in
this context. We have revised the wording to "**405 nm laser exposure within the ROI quantitatively
converted Cdh5-mClav at the junction (shown in magenta), whereas the other half of the ring remained
unconverted (shown in green)**" to ensure consistency with the description in the previous paragraph
(**lines 183-185**).

6. Rephrase line 267-268 to clarify that Rasip1 is within the apical surface in order to not imply that it
localises at the junctional rings.

Thank you for your comment. We have rephrased to clarify that Rasip1 is localized within the
presumed apical surface, ensuring it does not imply any localization at the junctional rings (**lines**

296-297).

7. Line 273: For broader readership, provide an explanation for what UCHD-mRuby labels since this
line is not introduced earlier.

We apologize for the oversight in not describing mRuby2-UCHD. In this study, we used mRuby2-
UCHD, which is the F-actin binding domain of utrophin fused to mRuby2, to label junction-associated
actin (lines 305-306).

Reviewer #4 (Remarks to the Author):

I co-reviewed this manuscript with one of the reviewers who provided the listed reports. This is part
of the Nature Communications initiative to facilitate training in peer review and to provide appropriate
recognition for Early Career Researchers who co-review manuscripts.

Thank you for informing us about the Nature Communications initiative and the role of co-reviewers
in this process. We appreciate your constructive feedback.

**References:**

Béraud-Dufour, S., Gautier, R., Albiges-Rizo, C., Chardin, P., & Faurobert, E. (2007). Krit 1

interactions with microtubules and membranes are regulated by Rap1 and integrin

cytoplasmic domain associated protein-1. *The FEBS Journal*, 274(21), 5518–5532.

<https://doi.org/https://doi.org/10.1111/j.1742-4658.2007.06068.x>

de Kreuk, B.-J., Gingras, A. R., Knight, J. D. R., Liu, J. J., Gingras, A.-C., & Ginsberg, M. H. (2016).

Heart of glass anchors Rasip1 at endothelial cell-cell junctions to support vascular integrity.

*ELife*, 5, e11394. <https://doi.org/10.7554/eLife.11394>

Gingras, A. R., Liu, J. J., & Ginsberg, M. H. (2012). Structural basis of the junctional anchorage of

the cerebral cavernous malformations complex. *Journal of Cell Biology*, 199(1), 39–48.

<https://doi.org/10.1083/jcb.201205109>

Glading, A., Han, J., Stockton, R. A., & Ginsberg, M. H. (2007). KRIT-1/CCM1 is a Rap1 effector

that regulates endothelial cell–cell junctions. *Journal of Cell Biology*, 179(2), 247–254.

<https://doi.org/10.1083/jcb.200705175>

Lampugnani, M. G., Orsenigo, F., Rudini, N., Maddaluno, L., Boulday, G., Chapon, F., & Dejana, E.

(2010). CCM1 regulates vascular-lumen organization by inducing endothelial polarity.

*Journal of Cell Science*, 123(7), 1073–1080. <https://doi.org/10.1242/jcs.059329>

Liu, J. J., Stockton, R. A., Gingras, A. R., Ablooglu, A. J., Han, J., Bobkov, A. A., & Ginsberg, M. H.

(2011). A mechanism of Rap1-induced stabilization of endothelial cell–cell junctions.

*Molecular Biology of the Cell*, 22(14), 2509–2519. [https://doi.org/10.1091/mbc.e11-02-](https://doi.org/10.1091/mbc.e11-02-0157)

0157

Sauter, L., Affolter, M., & Belting, H.-G. (2017). Distinct and redundant functions of Esam and

VE-cadherin during vascular morphogenesis. *Development*, dev-140038.

<https://doi.org/10.1242/dev.140038>

Sauter, L., Krudewig, A., Herwig, L., Ehrenfeuchter, N., Lenard, A., Affolter, M., & Belting, H.-G.

(2014). Cdh5/VE-cadherin Promotes Endothelial Cell Interface Elongation via Cortical Actin

Polymerization during Angiogenic Sprouting. *Cell Reports*, 9(2), 504–513.

<https://doi.org/https://doi.org/10.1016/j.celrep.2014.09.024>

Yin, J., Maggi, L., Wiesner, C., Affolter, M., & Belting, H.-G. (2024). Oscillatory contractile forces
refine endothelial cell-cell interactions for continuous lumen formation governed by
Heg1/Ccm1. *Angiogenesis*. <https://doi.org/10.1007/s10456-024-09945-5>

Reviewer #1 (Remarks to the Author):

The revisions have addressed my previous concerns. I have no further concerns. This is an excellent paper!

Thank you very much for your positive feedback and for your thoughtful review throughout the process. We are grateful for your comments, which helped improve the manuscript. We're glad to hear that you find the paper excellent.

Reviewer #2 (Remarks to the Author):

I enjoyed this manuscript and the careful revisions by the authors. The authors addressed all my comments from their first submission and I think this paper should be accepted for publication.

Thank you for your kind feedback and for taking the time to review our manuscript. We greatly appreciate your constructive comments and are glad that the revisions addressed your concerns. We're grateful for your recommendation for publication.

Reviewer #3 (Remarks to the Author):

The authors have significantly improved the manuscript by adding high-resolution images of immunofluorescent stainings, which now convincingly support the live imaging data. Additionally, they have made an effort to provide more rigorous and transparent quantitative data. I have only a few remaining minor comments:

Thank you for your positive feedback and for acknowledging the improvements in the manuscript. We are pleased that the high-resolution images and the additional quantitative data have strengthened the findings. We appreciate your thoughtful review and will carefully address the remaining minor comments below.

1. The authors should avoid using subjective language that overstates their findings.

Examples include:

a. line 21, 77: 'a wealth of molecular reporters', simply stating novel reporters or a variety of reporters would be sufficient

Thank you for your careful review and valuable suggestion. We have revised the text accordingly:

a. In line 21 and 77, we have replaced 'a wealth of molecular reporters' with "a series of".

b. line 80: '... advanced live imaging', what is defined as advanced?

Thank you for pointing this out. To clarify the term 'advanced live imaging' in line 80, we have revised the sentence to specify the techniques used, stating: "...using high-resolution and

multi-dimensional live imaging". We appreciate your suggestion to enhance the precision of our language.

c. Line 281: '... the massive detachment of cdh5...'

Thank you for your feedback on this phrasing. We have revised line 281 to remove the term 'massive' and replaced it with 'significant detachment of cdh5...'. We appreciate your suggestion to improve the clarity of our manuscript.

2. Line 135: there are no blue arrows in Fig 2A II

Thank you for bringing this to our attention. We have reviewed Figure 2A II and noticed that the blue arrows were mistakenly referenced. We have corrected the text, figure and figure legend to accurately reflect the figure.

3. Line 136-138: The statement on newly forming and existing apical membranes between tip cells and stalk cells adds essential information, but it would be helpful to also annotate these in Fig. 2B

Thank you for your valuable suggestion. We have added the necessary annotations to the figure to illustrate these distinctions more clearly. We appreciate your input in improving the manuscript.

4. Line 139: It remains unclear what threshold/cut off was used to define boundary area vs apical area in the quantifications of Fig 2B and C. Is this a fixed length from the periphery or based on an intensity threshold?

The boundary selection process involves a hybrid approach that combines automated analysis with manual input from the user. Users interact with the system through a graphical interface, where they can manually select and adjust boundaries using drawing tools. The boundary region is manually selected by the users. After that, the quantifications were performed using a fixed length of 0.5 μm from the periphery, which corresponds to approximately 2-3 pixels in the images. Proper description has been added to the Method section.

5. Line 366-371: A small schematic depicting the pathway would help in understanding the paragraph.

Thank you for your suggestion. We have included a small schematic in the revised manuscript to illustrate the pathway more clearly as **current Figure 8A**.

Reviewer #4 (Remarks to the Author):

Thank you for your contribution as a co-reviewer for this manuscript.